# Plan and Develop Advanced Knowledge and Skills for Future Industrial Employees in the Field of Artificial Intelligence, Internet of Things and Edge Computing

Łukasz Paśko [1], Maksymilian Mądziel [1], Dorota Stadnicka [1], Grzegorz Dec [2], Anna Carreras-Coch [3], Xavier Solé-Beteta [3], Lamprini Pappa [4], Chrysostomos Stylios [4,5,*], Daniele Mazzei [6] and Daniele Atzeni [6]

[1] Faculty of Mechanical Engineering and Aeronautics, Rzeszów University of Technology, 35-959 Rzeszów, Poland; lpasko@prz.edu.pl (Ł.P.); mmadziel@prz.edu.pl (M.M.); dorota.stadnicka@prz.edu.pl (D.S.)
[2] Faculty of Electrical and Computer Engineering, Rzeszów University of Technology, 35-959 Rzeszów, Poland; gdec@prz.edu.pl
[3] La Salle Campus Barcelona—Research Group in Internet Technologies & Storage, Universitat Ramon Llull, 08022 Barcelona, Spain; anna.carreras@salle.url.edu (A.C.-C.); xavier.sole@salle.url.edu (X.S.-B.)
[4] Laboratory of Knowledge and Intelligent Computing, Department of Informatics and Telecommunications, GR-47100 Arta, Greece; pappa.l@kic.uoi.gr
[5] Athena Research Center, Industrial Systems Institute, GR-26504 Patras, Greece
[6] Computer Science Department, University of Pisa, 56127 Pisa, Italy; daniele.mazzei@unipi.it (D.M.); daniele.atzeni@phd.unipi.it (D.A.)
* Correspondence: stylios@isi.gr or stylios@uoi.gr

**Abstract:** Knowledge and skills in the field of Artificial Intelligence (AI), Internet of Things (IoT), and Edge Computing (EC) are more and more important for industry. Therefore, it is crucial to know what current students and future employees can offer to the industry. University students develop their knowledge and skills to support the industry in implementing modern technologies in the future. It can be expected that the first source of information for students will be lectures and other activities at the university. However, they may obtain knowledge from other sources. This article presents the results of research conducted among students assessing their own knowledge and skills in the field of IoT, AI, and EC. The research was preceded by an analysis of curricula at selected universities in terms of topics related to AI, IoT, and EC. Based on the results of the analysis, survey questions were prepared. The developed questionnaire was made available to students. The research sample for the survey participants was 563 students. The results obtained were analyzed. The results of the analysis show which issues are better known to students and which are worse. The information presented in this paper can be a source of information for the industry that can assess the competences that are or will be available on the labor market in the near future. Additionally, universities can obtain information on the areas in which there are competency gaps and which methods of teaching AI, IoT, and EC are better perceived by students.

**Keywords:** Internet of Things; artificial intelligence; edge computing; knowledge; skills; university curricula

## 1. Introduction

Many industry forecasts for the Artificial Intelligence (AI) market can be found in the literature, and all agree on the importance and potential of this technology for increasing productivity in many industrial sectors. The dependence on the emergence of AI in running industries and shaping the education, transports, and health sectors is now well known in the literature [1]. For instance, Gartner has stated that "Worldwide AI software revenue is forecast to total $62.5 billion in 2022, an increase of 21.3% from 2021", and that "by 2025, the market for AI software will reach almost $134.8 billion" [2]. Within the same context, a

2021 study [3] highlighted the fact that the increasing use of AI in marketing can facilitate, expand, and modify markets themselves.

In the same line, the Internet of Things (IoT) is another key technology for the future of industry, according to all the literature. The IoT has been a key element for smart manufacturing, smart cities, smart health, smart grids, and electric vehicles, for example. IoT and Industrial Internet of Things (IIoT) bridges physical artifacts and the Internet, both in our daily lives and in the industry environment [4]. In this case, Gartner predicts that "The IoT-enabled applications and infrastructure software market will represent a 33-billion-dollar opportunity in 2025, up at a 34% Compound Annual Growth Rate (CAGR) from 2020. The largest revenue segment will be in IoT platforms, while the highest growth will occur in Customer Relationship Management (CRM)" [5].

Finally, Edge Computing (EC), being a much newer technology than the former two, is still also confirmed as one of the key technologies for the future of industry. Indeed, the combination of EC with AI and/or IoT is essential to reach the full potential of each technology separately. Gartner foresees that "by 2025, 30% of new industrial control systems will include analytics and AI-edge inference capabilities, up from less than 5% in 2021" [6].

In addition, previous studies have shown a positive relationship between technologies such as IoT or EC and the sustainable development goals, as described in [7]. The technologies considered may influence the achievement of selected goals.

Based on the previous industry forecasts, and probably also on other financial reports identifying China and the United States as the current leading countries in these markets, the European Commission has taken a number of relevant actions aiming to boost research and industrial capacity (while ensuring fundamental rights) in these areas.

In September 2020, a workshop organized by the Next Generation Internet of Things (NGIoT) project was the starting point for discussions on a next-generation IoT strategy. The workshop focused on Cloud-to-Edge-to-IoT and highlighted the need for an open industrial platform for cloud–edge orchestration, addressing the technology challenges and competitive impact for European stakeholders in light of their role in a data economy. Later, in March 2021, the Commission's Fireside Chat workshop mobilized and connected a small number of expert stakeholders from sectors such as aerospace, agriculture, automotive, construction machinery, home, and industrial automation. The aim of the workshop was to advance the design of a European strategy for the future IoT and EC, with a market window of over five years. In April 2021, based on the results of the Fireside Chat, the NGIoT and Edge Computing Strategy Forum (co-organized by the European Commission with the EU-IoT project) gathered technology experts from various digital and vertical domains to exchange views on the priorities, challenges, and opportunities ahead, and established a commonly shared strategic vision for the next-generation IoT and (far) EC in Europe.

Regarding funding programs, Horizon Europe is contributing more than EUR 150 billion into R&I under its 2021–2022 calls on "World Leading Data and Computing Technologies: From Cloud to Edge to IoT for European Data". In addition to Horizon Europe, the Commission's Digital Europe program (DIGITAL) will bring data and cloud services to EU businesses, citizens, and public administrations, aiming to set up common European data spaces in different verticals. DIGITAL will also strengthen existing AI testing and experimentation facilities in areas such as healthcare and mobility, and support the establishment of European digital innovation hubs in order to boost the digitization of small and medium-sized enterprises (SMEs) and help European companies become more competitive in the digital age. Furthermore, the EU, through the Digitizing European Industry (DEI), supports the development of industrial IoT platforms that are essential for the integration of key digital technologies in real-world applications, processes, products, and services. Complementing various policy initiatives, the Commission made around EUR 400 million available through its Horizon 2020 program for efforts on platform building and large-scale piloting under the DEI initiative. Regarding AI, the Commission plans to

invest EUR 1 billion per year through the Digital Europe and Horizon Europe programs and to mobilize additional investments from the private sector and the Member States in order to reach an annual investment volume of EUR 20 billion for AI over the course of the digital decade. Finally, the newly adopted Recovery and Resilience Facility has made EUR 134 billion available for digital. This will be a game-changer, allowing Europe to amplify its ambitions and become a global leader in developing cutting-edge and trustworthy AI [8].

One of the major concerns from an industry perspective is the lack of experts in the application of IoT, AI, and EC for their industries. Indeed, university course enrollment in AI and Machine Learning (ML) is increasing all over the world [9]. This means that applications of AI and ML have tremendous attraction and it is time for academies to adapt their educational offer to make AI and ML available to companies for answering Industry 4.0 (I4.0) challenges. It is universities' responsibility to train a new generation of experts in the application of "AI on the edge", starting from the actual companies' needs and creating a new paradigm for this. With this objective, the University of Pisa, University of Ioannina, Rzeszów University of Technology, and University Ramon Llull joined their efforts within the Planet4 project [10] (EC-funded under the Erasmus+ Knowledge Alliances program).

The Planet4 project aims to fill the gap between scientific research on AI and ML and its industrial application as enabling technology for the I4.0 paradigm. In the first stage of the project, a deep analysis of the knowledge and skills of future industrial employees in the field of AI, IoT, and EC was carried out. It was clear that, to fill the previous gap, the needs from industry had to be carefully collected [11], and that the contents/knowledge provided by universities had to be thoroughly analyzed [12]. However, Planet4 wanted to go even further, collecting the "real" knowledge acquired by the students.

This paper presents the work conducted within the Planet4 project. The objective of the study was to gather and analyze the knowledge and skills on AI, IoT, and EC acquired by the students as future industrial employees deploying such technologies. The main contribution of this work is its provision of knowledge about the declared students' competences in the field of AI, IoT, and EC. Many interesting conclusions arose from the survey, such as, for example, which particular areas of AI, IoT, and EC are least known by students, their need for other interdisciplinary knowledge for designing their technological developments, or even more subjective information such as what is harder for them to understand from the current curricula, or which are the most effective teaching methodologies for these areas from their point of view, etc.

Discussing the relevance of this study to the needs, it can be said that students' answers with the presented analysis are essential to fill the gap between industry needs and knowledge provided by academia regarding AI, IoT, and EC technologies. The data collected in the research brings information to academics that can be used in the development of new educational content and/or for the modification of existing courses of studies.

The rest of the paper is structured as follows: Section 2 reviews the literature on students' knowledge on AI, EC, and IoT for I4.0, Section 3 describes the methodology used for the analysis presented in this paper, Section 4 includes a summary of the curricula review, Section 5 details the questionnaires' implementation, Sections 6 and 7 correspond to the conduction and results of the survey, respectively, Section 8 discusses the previous results, and finally Section 9 concludes the paper.

## 2. Literature Review

The analysis of students' knowledge and the real success of a teaching process have always gone hand-in-hand with the development of effective courses and materials. However, this time-consuming process is particularly difficult to accomplish when dealing with research fields that are developing at a rapid pace, such as the science and technology field in recent decades.

The rapid development of research areas that have aroused great interest in the scientific community usually causes the academic world to focus on its pure research, pushing the state of the art to the limit, gradually moving away from the real needs of

the world of work. The latter cannot keep up with the latest research and studies, since the objectives and focus of these two words are fundamentally different. This distance is also reflected in academic teaching, causing a continuous estrangement between university students and their future jobs. For these reasons, at a certain point in the lifespan of a research field that can also lead to saleable applications and the creation of products, it has always been considered necessary to analyze the students' knowledge. Using this analysis, academics and companies are able to bridge the gap between the skills and notions taught in schools and universities and the necessary ones, synchronizing these two universes that, when they move at an important pace, tend to move apart and on parallel tracks.

This issue appeared in the last century, giving rise to new teaching techniques, such as problem-based learning [13]. Initially used mainly in the medical field [14], this technique aims to bridge the gap between theory and practice through complex real-world problems. Students learn to generalize concepts and principles without losing sight of the real-world application of what they are learning. Problem-based learning can also train skills that are underdeveloped by traditional teaching, such as critical thinking skills, problem-solving abilities, and communication skills.

At the same time as new teaching techniques, studies aimed at analyzing the knowledge and skill gaps have focused on knowledge transfer [15,16], i.e., "the process through which one unit (e.g., group, department, or division) is affected by the experience of another" [17]. The interest in bridging this gap is not limited to the medical and anthropological fields, but also spreads within other research areas, such as economics [18], chemistry [19], and engineering [20].

One of the fields that have attracted the most interest within academia and beyond in recent years, and thus is most likely to fall victim to the distancing effects between students and workers, is technology. Currently, the technological field is moving at an impressive speed, increasingly permeating our lives. AI has reached unthinkable milestones, advancing scientific research in computer vision [21], natural language processing [22], biology [23], etc. EC is developing more and more, especially in cooperation with the advent of IoT, and many studies are convinced that sooner or later it will be an indissoluble part of our daily life [24,25].

Therefore, it is not surprising that filling the knowledge gap has been necessary and has affected this area as well. A variety of studies have followed over the years, covering a wide range of areas, for example, the knowledge of graduate students in Computer Science or Software Engineering [26], the skills required by specific industries, such as robotics [27], or possible open-source software with which to address this gap [28].

Speaking of technological development and scientific fields in rapid development, we cannot fail to mention the so-called fourth industrial revolution. The term was coined by Henning Kagermann, Wolf-Dieter Lukas, and Wolfgang Wahlster in 2011. I4.0 refers to the inclusion of new technologies, such as AI and augmented reality, within the production process. Being a point of connection of the previously mentioned technological innovations and a paradigm that promises to revolutionize a large part of the industrial professional figures, it is easy to imagine the interest within this sector in the development of students and future workers equipped with an appropriate set of skills.

In this regard, Motyl et al. [29] surveyed nearly 500 students with the goal of understanding whether Italian engineering students are ready for I4.0. In their work, they detected the need to create educational courses able to give students a more complete understanding of the basic concepts of the current industrial revolution. Another survey questionnaire was used in [30] for collecting primary data in a study on the relationship between education and I4.0. In this article, the authors propose a new conceptual framework to be used in educational institutes in order to fill the observed gap between the skills required by Information Technology (IT) and manufacturing companies and the students' competences. In [31], the author, motivated by the lack of innovative teaching methods for AI on the Edge for I4.0 and the awareness of universities with respect to industrial needs, presents a new educational activity. This project is developed in four phases and aims to

create new technicians who are able to use I4.0-enabling technologies to solve real-world problems. A sustainable educational engineer archetype is proposed in the work [32]. The authors of the work recommend a way to introduce I4.0 technologies in educational programs to enhance awareness of sustainability in the industry. In [33], Matt et al. proposed a five-step methodological guide for introducing theoretical innovations related to I4.0 within SMEs. They highlighted the importance of knowledge transfer using both technical knowledge and best-practice examples and the need to change the roles, the competencies, and the skills of operators. They also validated their methodology using two different SMEs as case studies. Finally, in [34], the authors analyzed the skills and knowledge gap in the construction industry sector of a fast-developing country through a structured survey. The study shows an overall awareness of the new technologies among workers, but also reveals a low level of training preparation towards the adoption of these technologies, especially regarding human–machine communication, data analytics, and cyber security.

Our work goes beyond the SoA in many aspects. It integrates two important, and up to now separated, fields of research, namely students' knowledge and Industry 4.0. As such, it addresses both the skill gap between students and companies and the learning methodologies being used. Second, the scope of the work (often limited to one country) extends to all of Europe, thanks to the participation of various Universities of different countries. Third, it includes Master students, who are much closer to the labor market than Bachelor students, that were not considered in other reviews because they are less accessible and lower in amount. Fourth, it focuses on the latest Industry 4.0 technologies such as AI, IoT, and EC, including their integrated use/knowledge. Finally, the outcomes of our work are addressed through the production of new and innovative courses within the Planet4 project in the next year.

### 3. Work Methodology

This section presents the research methodology, which consists of five steps (see Figure 1). In the first step, the curricula of four universities (University of Pisa, University of Ioannina, Rzeszów University of Technology, University Ramon Llull) were reviewed. The researchers were looking for the courses in which the topics related to AI, IoT, and EC are discussed. The following information was collected: university, the name of the course, the name of the learning module, educational level, Intended Learning Outcomes (ILO), Teaching and Learning Activities (TLA), infrastructure indicated in curricula, software indicated in curricula, as well as teaching methods and techniques used in the didactic process. The main goal of this review was to identify the topics that are covered in university courses, the tools, software, environments, and the learning methods and techniques that are being used. To sum up, the aim was to determine the competencies that students should have in the areas analyzed. The results of the curricula review are discussed in Section 4.

The knowledge and competences described in the curricula were the basis for the development of the questionnaire, which is the second step of the methodology. Information on the identified topics, applications, software, tools, and methods of education was used to define the survey questions. The survey questions were divided into three parts related to the three topics under consideration: AI, IoT, and EC. The survey questions concerned, inter alia, students' knowledge about the areas and topics of AI, IoT, and EC that appeared in the curricula, familiarity with appropriate tools and software, knowledge of practical applications, and the suitability of teaching techniques used in universities.

The initial version of the survey was tested by a group of several dozen students. Each of them filled in the questionnaire and made some comments. Based on the results of the initial questionnaire and the comments from the students who answered the survey questions, the final version of the questionnaire was established. Detailed information on creating the questionnaires is provided in Section 5.

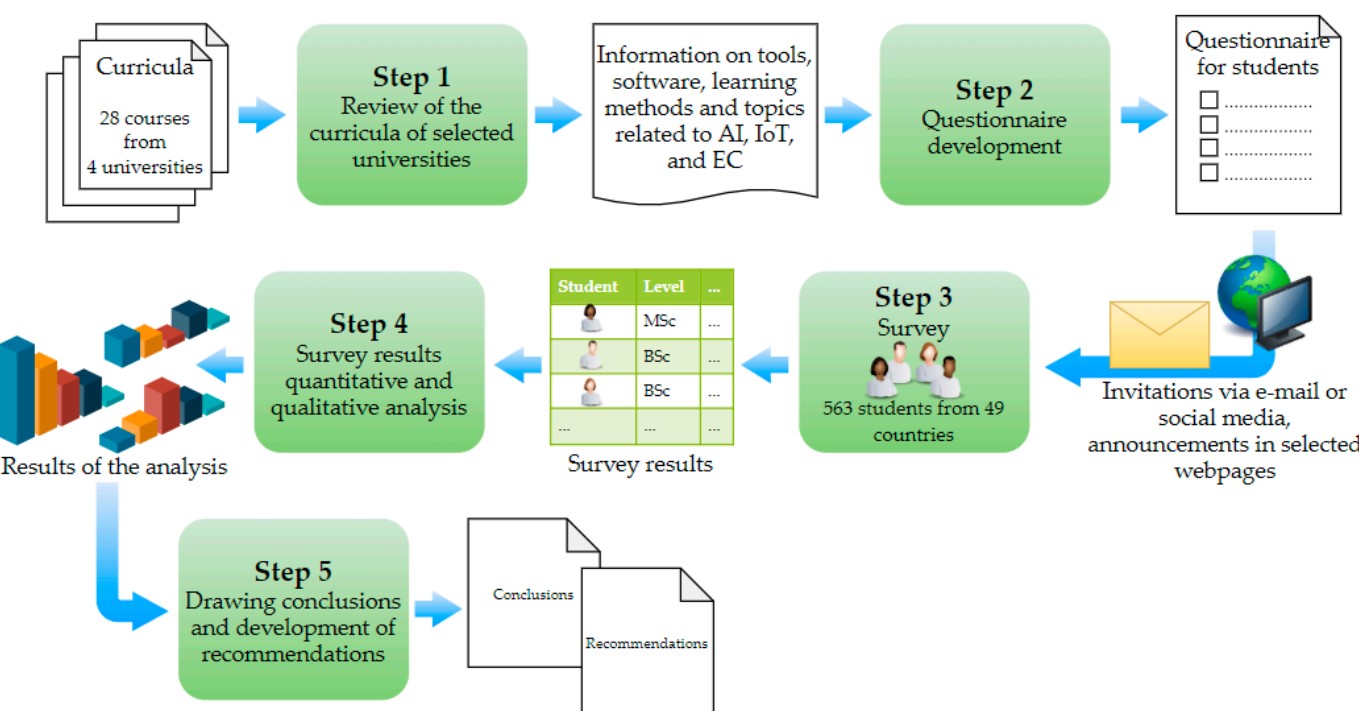

**Figure 1.** Work methodology.

The final version of the questionnaire was used to conduct the main survey (third step). Information about the survey was published on selected websites (including websites of universities and the Planet4 project). The questionnaire was also sent directly to several hundred students from dozens of countries by contacting them via e-mail or social media messages. Five hundred and sixty-three students from forty-nine countries responded positively to the invitation to participate in the survey. More information about the survey participants is provided in Section 6.

After the survey was completed, the answers to all survey questions were quantitatively and qualitatively analyzed (fourth step). The quantitative analysis included a summary of the number of answers in each level of the Likert scale for every close-ended question. It allowed to determine, inter alia, which AI, IoT, and EC topics are better known to students. In addition, the AI knowledge index was calculated to investigate the relationship between students' participation in projects and their knowledge in the field of AI. In turn, the qualitative analysis of students' answers took into account open-ended questions in which students could indicate what, in their opinion, is the most difficult in learning AI/IoT/EC, what are the shortcomings in the education process, and what could support them in learning AI/IoT/EC. The answers to all survey questions are summarized in Section 7.

The fifth step of the work methodology is based on the results of the analysis of students' surveys. The results of the analysis led to the formulation of conclusions about the students' knowledge and skills that they gain during their studies. The results also contributed to the proposal of some recommendations on the didactic process and changes in the content of university courses. This information is provided in Section 8.

## 4. Summary of Curricula Review

At the beginning of the study, the researchers studied the curricula. The curricula were analyzed at four universities participating in the Planet4 consortium. The researchers were particularly interested in the curricula of the faculties, which educate students in fields related mainly to computer science, industrial engineering, and mechanical engineering. Among the curricula analyzed, the focus was on those that are related to AI, IoT, or EC

at least to a minimal extent. The curricula were analyzed to find topics related to AI, IoT, or EC. The presence of at least one of these topics was an entry condition for an in-depth analysis—if at least one of the topics was discussed in a given curriculum, then that curriculum was transferred to a deeper analysis. In-depth analysis was conducted to extract information on ILOs, TLAs, teaching methods and techniques, as well as infrastructure and software used in the didactic process. Table 1 presents information on the courses and learning modules reviewed during the in-depth analysis of the curricula in the four universities mentioned above.

**Table 1.** Overview of courses and learning modules.

| | |
|---|---|
| Universities | University of Pisa (Italy), University of Ioannina (Greece), Rzeszów University of Technology (Poland), University Ramon Llull (Spain) |
| Courses | Aeronautics and Space Technology; Artificial Intelligence; Artificial Intelligence Fundamentals; Big Data Engineering; Big Data Mining; Computational Intelligence; Computational Intelligence and Deep Learning; Computational Mathematics for Learning and Data Analysis; Computer engineering (2); Data Analysis with Classic Techniques and Machine Learning Techniques; Data Mining; Data Mining and Machine Learning; Data Science; Electronic Engineering; Human Language Technologies; Internet of Things; Internet of Things and Edge Computing; Machine Learning; Management and Production Engineering; Material Engineering; Technologies for the Digital Transformation of the Companies (MTTD); Mechanics and Mechanical Engineering; Mechatronics; Optimization Methods and Game Theory; Statistical Machine Learning; Supply Chain Management and Technology; Transportation; University Expert in Digital Transformation |
| Learning modules | Advanced robot controls; AI (2); Automatics and domotics; Basics of AI; Bigdata, Analytics and AI; Computer networks in materials engineering; Computer vision; Computing infrastructures; Data analysis and visualization; Data analysis with R and Python languages; Data analytics; Data center technologies; Data mining (2); Decision support systems; Diagnostics and supervision of machining systems; Diagnostics of mechatronic systems; Digital transformation of the company; Distributed architecture projects; Embedded systems; Engineering of exploitation of road transportation means; Enterprise management support; Expert systems in aviation; Information technologies for I4.0; Intelligent computer systems; Intelligent measuring systems (2); Interconnection of data networks; Knowledge based systems; Knowledge discovery and data mining; Knowledge management; Local area networks; Methods of AI, IoT, Mobility and process automation; Modeling of production processes; Monographic lecture; NC controlled machines; Networking and IT security; Numerical simulation of technological processes; On-board control systems; Optimization methods; Projects in robotics; Statistics; Storage technologies; Supply chain technology; Technical robotics; Technology optimization; Telematics in transportation |
| Education level | BSc, MSc |
| ILO | 153 Intended Learning Outcomes (ILO) connected with AI, IoT and EC were reviewed |
| TLA | Descriptions of 255 Teaching and Learning Activities (TLA) connected with the mentioned ILOs were reviewed |
| Infrastructure indicated in curricula | Amazon Web Services; Cloudera Hadoop (CDH); Google Colab; IBM university platform; Microsoft Azure; KNX; Robots (wheeled and manipulation): Fanuc, ABB, Kawasaki, UR3; Students' laptop; Virtual machine |
| Software indicated in curricula | Aitech Sphinx (DeTreex, DSS, PC-Shell); Apache Spark; Arduino SDK; Azure Databricks; Azure IoT Hub; Azure Machine Learning; Azure Stack Edge; BI Beans; BotSociety; BotEngine; C programming language; C++ programming language; CISCO Packet Tracer; CLOS; Code Composer Studio; Comarch BI; DialogFlow; Docker; Flume; Hadoop Hbase; Hadoop HDFS; Hadoop Map/Reduce; IBM Cloud Watson; Java; JESS; Kafka; Kali Linux; Maple; Matlab; Matlab Fuzzy Logic Toolbox; MATLAB Neural Network Toolbox; Metasploit; Microsoft Bot Framework; Microsoft Dynamics AX Business Intelligence; Microsoft SQL Server Analysis Services; MPI Software; Octave; Prolog; Python programming language; PyTorch; QlikSense; QlikView; R programming language; Robot Operating System; Sci-kit learn; Siemens Simatic; Sqoop; Spark; SPSS; Statistica Data Miner; Statistica Neural Networks; Tableau; Tanagra; TensorFlow; TwinCAT; WEKA |
| Teaching Methods and Techniques | Case Method (through real cases and knowledge pills, the students learn about the subject), lecture, lab, individual project, team project, problem-based learning, Master class |

The analysis covered bachelor-level (BSc) and master-level (MSc) courses. Twenty-eight courses were identified where at least one of the three topics (AI, IoT, EC) is discussed. It is obvious that the topics related to AI, IoT, and EC were discussed in courses such as Artificial Intelligence, Computational Intelligence and Deep Learning, Machine Learning, or Internet of Things and Edge Computing. However, the analysis of the curricula showed that topics related to AI, IoT, and EC were also present, to a greater or lesser extent, in

courses that were not intended to educate specialists in these three fields mentioned. Examples of such courses include Aeronautics and Space Technology, Electronic Engineering, Management and Production Engineering, Mechanics and Mechanical Engineering, or Transportation. In the courses mentioned above, AI was by far the most frequent topic of the three considered. This confirms that modern universities recognize the broad nature of AI and the possibility of its application in many areas. In turn, EC is definitely the least frequent in the analyzed curricula. This is probably due to the fact that EC is a relatively novel topic and is unfamiliar to the wider academic community. Therefore, the implementation of this topic in the education process is at a low level.

The conducted analysis showed that AI, IoT, and EC are discussed in learning modules included in the considered courses. Some of these modules relate directly to the three topics considered. Examples of such modules are: Basics of artificial intelligence; Information technologies for I4.0; Embedded systems. It is worth emphasizing that a large part of the learning modules is related to data analysis and data processing (e.g., Big Data, Analytics and AI, Data analysis and visualization, Data mining, Knowledge discovery and data mining, Knowledge management). This may indicate a great interest in extracting knowledge from data using AI and ML methods. However, the topics considered are also applicable to completely different learning modules that are not directly related to them. Such modules are as follows: Computer networks in materials engineering, Diagnostics and supervision of machining systems, Engineering of exploitation of road transportation means, Local area networks, Modeling of production processes, Numerical Control (NC) machines, Numerical simulation of technological processes. This proves that teachers provide students with the ability to apply elements of AI, IoT, and EC in a variety of contexts and application fields.

In the identified learning modules, the researchers particularly investigated ILOs and TLAs, as well as infrastructure and software used in the didactic process. Some ILOs have been formulated in a short and concise manner by teachers (e.g., "Can operate the software to simulate artificial neural networks", "Has knowledge of selected modern optimization methods", "Explains fundamental ideas of I4.0"). However, most of the ILOs were more detailed and elaborate in their descriptions of the knowledge, skills, and competences of students after completing the education process. Some of the ILOs that came from the Management and Production Engineering course are listed below.

- Knows contemporary tools of AI, including artificial neural networks and genetic algorithms, and can use them to solve complex tasks and problems occurring in management and production engineering;
- Knows modern information technologies, such as Online Analytical Processing (OLAP), data warehouses, methods and tools of AI, and can use them to create intelligent decision systems;
- Knows the methods and tools of AI and can use them to create knowledge bases to support the knowledge management process;
- Knows the basic statistical methods and advanced methods of AI, necessary for the analysis of engineering, business, or production data, and is able to use them to solve tasks;
- Has the ability to use appropriate software to solve specific decision problems, both single-criteria and multi-criteria, and to create advisory systems using the MATLAB Fuzzy Logic Toolbox software for problems occurring in conditions of uncertainty.

In the case of TLAs, teachers described the content of education discussed in class, characterized the issues debated with students, and pointed to the ways in which knowledge is transferred to students. A few examples of TLAs that apply to the Management and Production Engineering course are listed below:

- Fundamentals of neural networks. Biological bases of neurocomputing, basic model of neuron and neural network. Basic rules for teaching neural networks (supervised—the delta rule, and unsupervised—the Hebb rule), the concept of error function, the prob-

lem of generalization, the role of training and test set. Basic neural network learning algorithm—back propagation method, types of back propagation algorithms. Self-organizing neural networks (SONN): basics, neighborhood function, practical aspects of calculations using Self Organizing Maps (SOM). Neural networks with feedback: Hopfield and Hamming networks. Practical applications of neural networks for solving tasks: classification, clustering, forecasting, image processing, and recognition in automation.

- Application of AI methods: hybrid systems. Decision support system (DSS) based on the knowledge base—intelligent DSS. Design and implementation of intelligent DSS with the use of AI tools (neural networks, genetic algorithms, and fuzzy logic).
- Preparation of training data sets for modeling and simulating artificial neural networks in the Statistica Neural Networks software. Solving practical tasks of classification, forecasting, and grouping with the use of neural networks, including a multi-layer perceptron, radial basis function network and Kohonen neural network.

In many TLAs, teachers mentioned specific software and infrastructure used to conduct classes. In the case of infrastructure, the analyzed curricula showed examples of computing platforms necessary to conduct the classes (Amazon Web Services, Hadoop, Google Colab, IBM university platform, Microsoft Azure). Moreover, the teachers indicated the need to use appropriate computer equipment and robots (in some classes). Among the software included in the curricula, there are not only ready-made applications and software frameworks, but also programming languages (e.g., C, C ++, Java, Python, R). It is noteworthy that there is a large part of software designed for data analysis and processing.

The last issue searched for in the curricula was teaching methods and techniques. In the learning modules analyzed, lectures, labs, and projects (individual or team) were especially used. However, problem-based learning, case methods, and master classes were also involved in the teaching process.

The information mentioned above, which was identified during the analysis of the curricula, was used in the development of the survey questions and the preparation of the predefined response lists for close-ended questions.

## 5. Questionnaire Development

The first step in developing the survey was to establish its structure. It was assumed that the questionnaire would consist of the following four parts:

- General questions covering: the country or countries where the student is or was learning, level of study, and field of study;
- AI-related questions;
- IoT-related questions;
- EC-related questions.

Next, each part of the questionnaire was filled with the appropriate questions. The questions in the AI, IoT, and EC sections were related to:

- Topics that are connected with AI/IoT/EC;
- Degree of students' knowledge about tools/software/environment that can be used in AI/IoT/EC;
- Applications and contexts of using AI/IoT/EC;
- Learning techniques used in the education process;
- Difficulties in learning AI/IoT/EC;
- Students' needs associated with the learning process.

The questionnaire included both open-ended and close-ended questions. A list of predefined answers was assigned to each close-ended question. The predefined answers were based on an extended Likert scale with the following answers: not at all, to a small extent, to some extent, to a moderate extent, to a great extent, to a very great extent. An atypical six-point scale was used in the research. The answer "not at all" has been added to a five-point scale to clearly highlight the topics that the students do not know about. Thus,

the middle value ("to a moderate extent") has shifted to the right. This may have led the respondents to find this answer positive rather than mediocre. The results were analyzed, taking this into account. The basis for creating the list of predefined answers was primarily the review of the curricula.

The result of the tasks mentioned above was the preliminary questionnaire. On the basis of the preliminary questionnaire, the initial survey was conducted. During the initial survey, the questionnaires were distributed to a limited number of students. The aim of the initial survey was to collect feedback from students. The students not only answered the survey questions, but could also add comments to individual survey questions and write a comment on the entire survey. Many students took advantage of this opportunity. Due to this, it was possible to determine whether the survey questions were legible and understandable for students.

Based on the conclusions of the initial survey, some corrections were made to the questionnaire. First of all, some of the predefined answers of the close-ended questions have been modified or removed. In addition, the content of some questions was reformulated to better reflect their intended meaning. This is how the final version of the questionnaire was created. The final version was sent to students.

## 6. Conducting a Survey

The final questionnaire was distributed to students using two channels: named invitations sent to selected persons and public announcements on the Internet requesting to fill out the survey. The research was carried out from 8 March 2021 to 7 May 2021. The questionnaire was filled out by 563 students. In the research, 26 European countries were represented. In addition to European countries, the following countries were indicated by students: Afghanistan, Azerbaijan, Canada, China, Egypt, India, Iran, Kenya, Nigeria, Pakistan, Sri Lanka, USA, and Venezuela. The number of questionnaires by country is shown in Table 2.

**Table 2.** Countries represented in the survey research.

| Country | Number of Questionnaires | Country | Number of Questionnaires | Country | Number of Questionnaires |
|---|---|---|---|---|---|
| Afghanistan | 1 | Greece | 54 | Pakistan | 3 |
| Azerbaijan | 2 | India | 4 | Poland | 97 |
| Belgium | 2 | Iran | 3 | Portugal | 29 |
| Bulgaria | 1 | Ireland | 1 | Romania | 149 |
| Canada | 4 | Italy | 90 | Russia | 1 |
| China | 1 | Kenya | 3 | Serbia | 2 |
| Croatia | 2 | Latvia | 1 | Spain | 64 |
| Czechia | 1 | Liechtenstein | 1 | Sri Lanka | 1 |
| Denmark | 1 | Lithuania | 8 | Sweden | 6 |
| Egypt | 1 | Macedonia | 1 | Turkey | 5 |
| Finland | 1 | Netherlands | 4 | United Kingdom | 8 |
| France | 20 | Nigeria | 2 | United States | 2 |
| Germany | 15 | Norway | 1 | Venezuela | 1 |
| | | | | TOTAL | 593 |

The students who participated in the research studied at different study levels (Figure 2). Study levels were divided into three levels: bachelor, master, and Ph.D. Twenty-six students indicated a different level of studies than those mentioned above. The sum of the numbers in Figure 2 are greater than the number of questionnaires (563) because some students indicated more than one study level. The number of questionnaires obtained from students studying at the bachelor level was by far the largest. However, this does not mean that the survey was primarily dedicated to these students. The large number of bachelor-level questionnaires is related to the fact that it is the most accessible level of study with the highest number of students.

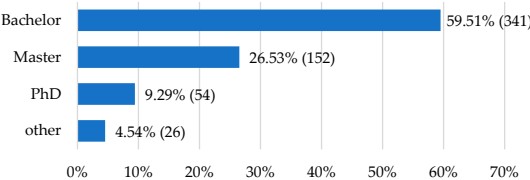

**Figure 2.** Students' study levels.

Students who took part in the survey studied in various fields of study (see Figure 3). The sum of the numbers in Figure 3 is greater than the number of questionnaires (563) as some students indicated more than one field of study. The highest percentage of students come from IT Engineering (41.72%). In addition, some respondents indicated fields of study highly related to IT Engineering, such as Computer Science (3.19%), Computer Engineering and Informatics (0.33%), Informatics and Computer Science (0.17%), Informatics (0.17%), or Electrical, Telecommunication, and Computer Science (0.17%) Information Systems (0.17%). The second most frequently mentioned field of study was Industrial Engineering (15.40%) and the third was Mechanical Engineering (8.94%). Each of the other fields of study represented in the surveys gave less than 4% of the questionnaires obtained.

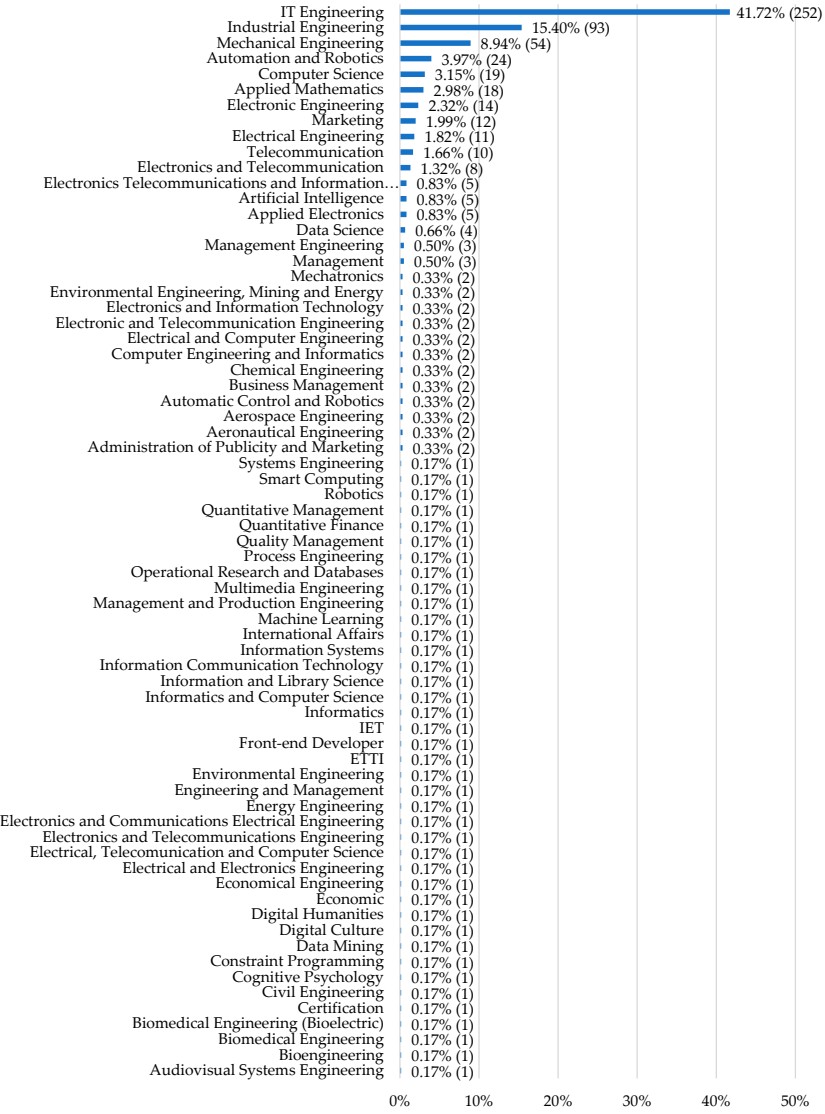

**Figure 3.** Students' fields of study.

After preliminary qualitative analysis of the survey results, two questionnaires were removed from the further analysis as they contained no answers (the questionnaires contained only information about the respondents). From this, it follows that 561 questionnaires were further analyzed.

## 7. Results of the Survey

The answers received from the students were reviewed to check their quality. Quantitative and qualitative analyses of the answers were performed. At the beginning, a general analysis of the answers obtained was carried out, taking into account all three areas of research (AI, IoT, EC). Next, each research area was individually analyzed. During the individual analysis of a given area, we only considered surveys in which students indicated that they were learning topics related to this area.

### 7.1. General Overview

In this section, how many students learned AI, IoT, and EC is presented. The total number of responses was 561. Table 3 shows that the largest number of respondents indicated that they learned AI (305 answers). The second place in terms of declared knowledge among students was IoT (139 answers). However, the number of AI students was more than double that of IoT students. By far, the lowest number of students declared knowledge of EC (23 answers). The highest popularity of AI among students may be due to the fact that AI-related issues are much more often present in the content of university courses compared to IoT and EC. This is confirmed by the analysis of the content of courses, performed before the survey.

**Table 3.** Number of students who declared learning in the three areas under consideration.

| Have You Ever Learned About? | AI | IoT | EC |
|---|---|---|---|
| Yes | 305<br>54.37% | 139<br>24.78% | 23<br>4.10% |
| No | 256<br>45.63% | 409<br>72.91% | 509<br>90.73% |
| No answer | 0<br>0.00% | 13<br>2.32% | 29<br>5.17% |

On the other hand, the lowest popularity of EC may result from the fact that it is a relatively new term and is not yet strongly present in universities as well as in business activities. The highest number of empty answers ("No answer") may also prove the low recognition of the term "edge computing".

From Figure 4, it can be seen that only 13 respondents indicated the answer "yes" to AI, IoT, and EC simultaneously, which is approximately 2.32%. This shows that a student who acquires knowledge and skills in all three areas will be able to boast of unique knowledge and can be a particularly valuable employee in the modern labor market. Among the AI respondents, 95 students also declared learning IoT, and 5 students indicated EC, but most of them declared knowledge of AI only (192).

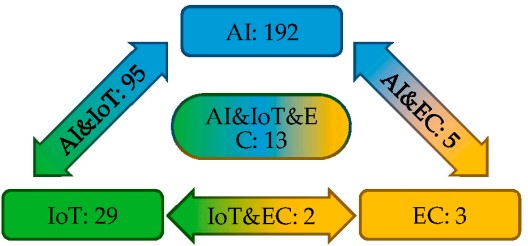

**Figure 4.** Number of students who declare knowledge of the three areas under consideration.

In the case of IoT, it is characteristic that most students combine knowledge of IoT with AI (95). Only 29 respondents declare knowledge of IoT without the other two areas. This shows how the two areas, IoT and AI, are related to each other. This may also indicate that the content of university courses brings these two areas together.

### 7.2. Artificial Intelligence

This section presents the level of students' knowledge in the area of AI. The percentages calculated in the tables below (from Tables 4–14 and also in Table 18) assume that 100% is the number of students who chose "yes" in the question "Have you ever learned about AI?".

Table 4 summarizes the data concerning AI, which was divided into seven areas in the survey. It can be seen that ML is the most recognized area of AI among students. A total of 96.72% of the students indicated at least a low level of knowledge of ML. More than a quarter of the respondents declared that they had no knowledge of natural language processing, computer vision, and cognitive computing. These three areas are the largest gaps in the education of students among the areas of AI mentioned in this question. The names of these areas indicate the practical applications of AI, such as natural language translation or image segmentation. This may mean the need to emphasize the practical applications of AI when teaching students. A particular need seems to be cognitive computing, which was unknown to more than 40% of the respondents.

**Table 4.** The level of students' knowledge in the AI areas considered.

| AI Area | Not at All | To a Small Extent | To Some Extent | To a Moderate Extent | To a Great Extent | To a Very Great Extent | ND |
|---|---|---|---|---|---|---|---|
| Machine learning | 9 <br> 2.95% | 59 <br> 19.34% | 93 <br> 30.49% | 68 <br> 22.30% | 54 <br> 17.70% | 21 <br> 6.89% | 1 <br> 0.33% |
| Deep learning | 34 <br> 11.15% | 88 <br> 28.85% | 74 <br> 24.26% | 46 <br> 15.08% | 40 <br> 13.11% | 14 <br> 4.59% | 9 <br> 2.95% |
| Data mining | 51 <br> 16.72% | 85 <br> 27.87% | 67 <br> 21.97% | 47 <br> 15.41% | 33 <br> 10.82% | 11 <br> 3.61% | 11 <br> 3.61% |
| Computation intelligence | 56 <br> 18.36% | 77 <br> 25.25% | 70 <br> 22.95% | 52 <br> 17.05% | 31 <br> 10.16% | 7 <br> 2.30% | 12 <br> 3.93% |
| Natural language processing | 82 <br> 26.89% | 85 <br> 27.87% | 51 <br> 16.72% | 35 <br> 11.48% | 32 <br> 10.49% | 9 <br> 2.95% | 11 <br> 3.61% |
| Computer vision | 77 <br> 25.25% | 86 <br> 28.20% | 62 <br> 20.33% | 37 <br> 12.13% | 24 <br> 7.87% | 9 <br> 2.95% | 10 <br> 3.28% |
| Cognitive computing | 126 <br> 41.31% | 93 <br> 30.49% | 42 <br> 13.77% | 16 <br> 5.25% | 14 <br> 4.59% | 3 <br> 0.98% | 11 <br> 3.61% |

In the previous question, students indicated that ML is the area of AI most familiar to them. Table 5 presents the level of declared students' knowledge in the field of ML techniques. It can be seen that the level of knowledge of supervised learning techniques is the highest—more than a quarter of the students indicated "to a great extent" or "to a very great extent" responses, and only about 10% of the students did not have knowledge of supervised learning. The reason for this may be that supervised learning methods are relatively simple compared to the others. More advanced techniques are less popular among students. Reinforcement learning turned out to be the greatest need in the field of ML. Almost 30% of the students had little knowledge about it and almost 28% had no knowledge about this topic at all.

**Table 5.** The level of students' knowledge in basic ML techniques.

| ML Technique | Not at All | To a Small Extent | To Some Extent | To a Moderate Extent | To a Great Extent | To a Very Great Extent | ND |
|---|---|---|---|---|---|---|---|
| Supervised learning | 33 10.82% | 75 24.59% | 52 17.05% | 62 20.33% | 48 15.74% | 32 10.49% | 3 0.98% |
| Semi-supervised learning | 52 17.05% | 92 30.16% | 58 19.02% | 51 16.72% | 31 10.16% | 8 2.62% | 13 4.26% |
| Unsupervised learning | 54 17.70% | 86 28.20% | 47 15.41% | 46 15.08% | 40 13.11% | 18 5.90% | 14 4.59% |
| Reinforcement learning | 85 27.87% | 90 29.51% | 48 15.74% | 36 11.80% | 25 8.20% | 6 1.97% | 15 4.92% |

Students' knowledge of deep learning was slightly lower compared to ML. Only 11.15% of the students were not familiar with deep learning. Table 6 provides a closer look at the main deep learning models in the context of students' knowledge. It can be seen that there is a particularly low level of knowledge regarding Generative Adversarial Network (GAN) and transformer. Almost 50% of the students did not know these models at all, and only about 10% knew of them at least to a moderate extent. Looking at the other two models, the most well-known is the convolutional neural network; however, over 22% did not know this model at all. The results show that the issues related to deep learning are clearly less recognizable than those related to ML. This may suggest the need to enrich the content of university courses with topics related to deep learning.

**Table 6.** The level of students' knowledge on the main deep learning models.

| Deep Learning Model | Not at All | To a Small Extent | To Some Extent | To a Moderate Extent | To a Great Extent | To a Very Great Extent | ND |
|---|---|---|---|---|---|---|---|
| Convolutional neural network | 68 22.30% | 70 22.95% | 53 17.38% | 50 16.39% | 38 12.46% | 22 7.21% | 4 1.31% |
| Recurrent neural network | 81 26.56% | 81 26.56% | 49 16.07% | 44 14.43% | 26 8.52% | 10 3.28% | 14 4.59% |
| Transformer | 145 47.54% | 70 22.95% | 38 12.46% | 23 7.54% | 11 3.61% | 4 1.31% | 14 4.59% |
| Generative adversarial network (GAN) | 152 49.84% | 72 23.61% | 25 8.20% | 21 6.89% | 16 5.25% | 4 1.31% | 15 4.92% |

In the question about AI areas, students were asked, among others, about their knowledge of data mining. A total of 79.67% of the respondents answered that they had any knowledge of this subject, and 29.84% of the students rated this knowledge to be at least of moderate extent. Table 7 presents the declared knowledge in the field of data mining divided into six phases of the Cross Industry Standard Process for Data Mining (CRISP-DM). It is characteristic that students rate their knowledge to be the worst in the first and last phases of CRISP-DM. The four internal phases of the CRISP-DM that are related to data understanding and processing, modeling, and evaluation, are rated better. These results indicate the need to familiarize students with the business context of the analyzed data. This also draws attention to the implementation issues of the models developed in the business field. Data processing and modeling are not enough to master the entire data mining process and make it an added value in business applications. Without business understanding and deployment phases, even the best work of data analysts can be wasted.

**Table 7.** The level of students' knowledge in data mining phases (CRISP-DM).

| Data Mining Phase | Not at All | To a Small Extent | To Some Extent | To a Moderate Extent | To a Great Extent | To a Very Great Extent | ND |
|---|---|---|---|---|---|---|---|
| Business understanding | 115 37.70% | 72 23.61% | 60 19.67% | 23 7.54% | 19 6.23% | 8 2.62% | 8 2.62% |
| Data understanding | 76 24.92% | 65 21.31% | 52 17.05% | 48 15.74% | 36 11.80% | 10 3.28% | 18 5.90% |
| Data preparation | 77 25.25% | 59 19.34% | 56 18.36% | 41 13.44% | 31 10.16% | 21 6.89% | 20 6.56% |
| Modeling | 85 27.87% | 55 18.03% | 57 18.69% | 43 14.10% | 29 9.51% | 21 6.89% | 15 4.92% |
| Evaluation | 89 29.18% | 66 21.64% | 40 13.11% | 45 14.75% | 28 9.18% | 21 6.89% | 16 5.25% |
| Deployment | 105 34.43% | 81 26.56% | 43 14.10% | 35 11.48% | 16 5.25% | 10 3.28% | 15 4.92% |

The next question in the survey concerns computation intelligence. In the first question, 77.70% of the students indicated that they had any knowledge of this area of AI. Table 8 summarizes the data concerning three aspects of computation intelligence. It can be seen that most of the students are familiar with neural networks. About 20% of the respondents considered their level of knowledge of neural networks to be high or very high. The situation is worse in the other two aspects. In particular, fuzzy logic seems to be a field that requires more attention in the didactic process because more than a third of students declared no knowledge of fuzzy logic at all.

**Table 8.** The level of students' knowledge in computation intelligence aspects.

| Computation Intelligence Aspect | Not at All | To a Small Extent | To Some Extent | To a Moderate Extent | To a Great Extent | To a Very Great Extent | ND |
|---|---|---|---|---|---|---|---|
| Fuzzy systems | 106 34.75% | 70 22.95% | 57 18.69% | 42 13.77% | 18 5.90% | 3 0.98% | 9 2.95% |
| Neural networks | 34 11.15% | 65 21.31% | 69 22.62% | 57 18.69% | 42 13.77% | 23 7.54% | 15 4.92% |
| Genetic algorithms | 76 24.92% | 83 27.21% | 49 16.07% | 51 16.72% | 20 6.56% | 9 2.95% | 17 5.57% |

The next question in the survey pointed to a more practical area of AI related to natural language processing. The results obtained in the first question show that this is one of the worst-known areas of AI. Table 9 breaks down this area into three aspects. Each of them seems to be very poorly known among students. It can be seen that 40% of the students did not know the issues related to natural language generation and natural language translation. The level of knowledge in the third aspect, speech recognition, is only slightly better.

Another practical area of AI is computer vision. As in the previous area, the results obtained in the first question show that the level of general knowledge is rather low. Table 10 breaks computer vision into five aspects. Among these aspects, the best known is image classification. It can be seen that image classification is known at least to a minimal extent by 75.74% of the respondents. On the other hand, the worst is the level of knowledge about domain adaptation and neural style transfer, which are unknown to more than 45% of respondents. These two aspects are the most important needs in the context of increasing computer vision competencies.

**Table 9.** The level of students' knowledge in natural language processing aspects.

| Natural Language Processing Aspect | Not at All | To a Small Extent | To Some Extent | To a Moderate Extent | To a Great Extent | To a Very Great Extent | ND |
|---|---|---|---|---|---|---|---|
| Speech recognition | 105 34.43% | 93 30.49% | 44 14.43% | 31 10.16% | 16 5.25% | 5 1.64% | 11 3.61% |
| Natural language generation | 121 39.67% | 67 21.97% | 48 15.74% | 27 8.85% | 12 3.93% | 7 2.30% | 23 7.54% |
| Natural language translation | 122 40.00% | 72 23.61% | 38 12.46% | 28 9.18% | 16 5.25% | 6 1.97% | 23 7.54% |

**Table 10.** The level of students' knowledge in computer vision aspects.

| Computer Vision Aspect | Not at All | To a Small Extent | To Some Extent | To a Moderate Extent | To a Great Extent | To a Very Great Extent | ND |
|---|---|---|---|---|---|---|---|
| Image classification | 63 20.66% | 70 22.95% | 63 20.66% | 42 13.77% | 36 11.80% | 20 6.56% | 11 3.61% |
| Object localization and detection | 70 22.95% | 84 27.54% | 56 18.36% | 42 13.77% | 21 6.89% | 13 4.26% | 19 6.23% |
| Image segmentation | 79 25.90% | 88 28.85% | 48 15.74% | 34 11.15% | 22 7.21% | 16 5.25% | 18 5.90% |
| Domain adaptation | 142 46.56% | 75 24.59% | 42 13.77% | 14 4.59% | 7 2.30% | 6 1.97% | 19 6.23% |
| Neural style transfer | 149 48.85% | 68 22.30% | 34 11.15% | 18 5.90% | 12 3.93% | 6 1.97% | 18 5.90% |

The last analyzed area of AI is cognitive computing. This area was the worst in terms of students' knowledge. In the first question, 41.31% of the respondents confirmed that they had no knowledge of cognitive computing. Table 11 confirms previous observations regarding the low level of students' knowledge in this area. Cognitive computing is divided into three aspects. It can be seen that the aspect related to meta-algorithms has by far the lowest level of students' knowledge. Most of the respondents (58.03%) have no knowledge of this topic. This shows that the entire area of cognitive computing may be a need in the context of broadening students' knowledge in the field of widely understood AI and its practical applications.

**Table 11.** The level of students' knowledge in cognitive computing aspects.

| Cognitive Computing Aspect | Not at All | To a Small Extent | To Some Extent | To a Moderate Extent | To A Great Extent | To a Very Great Extent | ND |
|---|---|---|---|---|---|---|---|
| Interactive task learning | 147 48.20% | 84 27.54% | 38 12.46% | 15 4.92% | 6 1.97% | 5 1.64% | 10 3.28% |
| Game playing agents | 135 44.26% | 66 21.64% | 51 16.72% | 20 6.56% | 7 2.30% | 6 1.97% | 20 6.56% |
| Meta-algorithms in cognitive computing | 177 58.03% | 59 19.34% | 27 8.85% | 11 3.61% | 5 1.64% | 6 1.97% | 20 6.56% |

The students also answered the question of what programming languages they know in AI applications. The level of students' knowledge about programming languages was assessed on the same scale as in the case of the questions discussed above. Table 12 summarizes the responses to this question. We can see two programming languages that stand out in terms of students' knowledge. They are Python and MATLAB. Python stands out in particular because 30.49% of respondents knew it to a high or very high degree. On

the opposite side are three languages that most of the respondents have never used in AI. They are R, Prolog, and Lisp.

**Table 12.** The level of students' knowledge in the following programming languages in AI applications.

| Programming Language | Not at All | To a Small Extent | To Some Extent | To a Moderate Extent | To a Great Extent | To a Very Great Extent | ND |
|---|---|---|---|---|---|---|---|
| C/C++ | 83 27.21% | 44 14.43% | 61 20.00% | 57 18.69% | 39 12.79% | 12 3.93% | 9 2.95% |
| Python | 49 16.07% | 48 15.74% | 47 15.41% | 54 17.70% | 58 19.02% | 35 11.48% | 14 4.59% |
| Lisp | 244 80.00% | 23 7.54% | 7 2.30% | 5 1.64% | 2 0.66% | 2 0.66% | 22 7.21% |
| Java | 118 38.69% | 57 18.69% | 35 11.48% | 33 10.82% | 27 8.85% | 17 5.57% | 18 5.90% |
| MATLAB | 59 19.34% | 60 19.67% | 72 23.61% | 53 17.38% | 35 11.48% | 12 3.93% | 14 4.59% |
| Prolog | 229 75.08% | 33 10.82% | 13 4.26% | 4 1.31% | 4 1.31% | 4 1.31% | 18 5.90% |
| R | 181 59.34% | 40 13.11% | 27 8.85% | 18 5.90% | 15 4.92% | 7 2.30% | 17 5.57% |

In the survey, the students had the opportunity to list other languages they use or have used in AI applications. Thirty-seven students indicated at least one additional programming language, as follows (the values in parentheses indicate the number of students pointing to a given language): JavaScript (7), C# (6), Julia (5), Scala (4), Java (3), CLIPS (2), OCaml (2), NET, Apex, Assembly, F#, GOAL, HALCON, Haskell, HTML, MDX, MiniZinc, PHP, Rust, Salesforce, SAS, SQL, Vivado, Webservice, XML.

The students were also asked about the software used in AI applications. Table 13 shows the level of students' knowledge of the eleven software or environments that appeared in the survey. It can be seen that, among the tools listed in this question, MATLAB was the best-known tool. A total of 70.16% of students who learned about AI knew this software. The least-known tools were AITECH SPHINX, Scilab, and SWI Prolog—more than 70% of the students did not know these tools at all.

As with the programming languages, the students could name other software that they use in AI. Fifty students indicated at least one software, as follows (the values in parentheses indicate the number of students pointing to a given software): Jupyter Notebook (18), Google Colab (14), JetBrains IntelliJ IDEA (3), Visual Studio (3), Atom (2), Eclipse (2), Emacs (2), Sublime Text (2), WEKA (2), Apache Netbeans, CLion, Code Blocks, Deepnote, Docker, GoLand, Julia, Knime, OpenVINO, Pandas, PyTorch, RapidMiner Studio, Scikit-learn, TensorFlow, Torch, Vim (Linux).

The high frequency of Google Colab and Jupyter Notebook indications may be due to the popularity of the Python language, as shown by one of the previous questions. These tools are often used by Python programmers.

The next question is about the students' knowledge of specific practical applications of AI. Table 14 presents the data for twelve selected AI applications. It can be seen that optimization issues were known to 64.26% of the students. In the case of the remaining eleven applications, the level of students' knowledge was lower—the number of students declaring knowledge of other applications did not exceed 60%. The fewest respondents could use AI in supply chains management—approximately 50% had never used AI in this area. In addition, 40 to 50% of the students had no knowledge of how to use AI for manufacturing processes monitoring, deliveries, scheduling problems, cognitive systems, and robots.

**Table 13.** The level of students' use of the software/environment in AI.

| Software/Environment | Not at All | To a Small Extent | To Some Extent | To a Moderate Extent | To a Great Extent | To a Very Great Extent | ND |
|---|---|---|---|---|---|---|---|
| AITECH SPHINX | 254 83.28% | 19 6.23% | 5 1.64% | 1 0.33% | 3 0.98% | 1 0.33% | 22 7.21% |
| Statistica | 211 69.18% | 36 11.80% | 15 4.92% | 7 2.30% | 5 1.64% | 3 0.98% | 28 9.18% |
| MATLAB | 72 23.61% | 64 20.98% | 57 18.69% | 50 16.39% | 28 9.18% | 15 4.92% | 19 6.23% |
| MS Excel | 118 38.69% | 44 14.43% | 31 10.16% | 40 13.11% | 27 8.85% | 18 5.90% | 27 8.85% |
| Scilab | 235 77.05% | 27 8.85% | 4 1.31% | 7 2.30% | 2 0.66% | 0 0.00% | 30 9.84% |
| RStudio | 186 60.98% | 42 13.77% | 17 5.57% | 13 4.26% | 12 3.93% | 7 2.30% | 28 9.18% |
| SWI Prolog | 234 76.72% | 20 6.56% | 8 2.62% | 6 1.97% | 6 1.97% | 1 0.33% | 30 9.84% |
| PyCharm | 130 42.62% | 40 13.11% | 25 8.20% | 35 11.48% | 33 10.82% | 18 5.90% | 24 7.87% |
| Spyder | 181 59.34% | 25 8.20% | 21 6.89% | 19 6.23% | 21 6.89% | 9 2.95% | 29 9.51% |
| Visual Studio Code | 101 33.11% | 55 18.03% | 30 9.84% | 40 13.11% | 28 9.18% | 31 10.16% | 20 6.56% |
| Anaconda | 122 40.00% | 35 11.48% | 21 6.89% | 33 10.82% | 37 12.13% | 35 11.48% | 22 7.21% |

**Table 14.** The level of students' knowledge about how to use AI in the following applications.

| AI Applications | Not at All | To a Small Extent | To Some Extent | To a Moderate Extent | To a Great Extent | To a Very Great Extent | ND |
|---|---|---|---|---|---|---|---|
| Quality problems | 104 34.10% | 73 23.93% | 53 17.38% | 29 9.51% | 17 5.57% | 7 2.30% | 22 7.21% |
| Predictive maintenance | 116 38.03% | 68 22.30% | 35 11.48% | 27 8.85% | 21 6.89% | 7 2.30% | 31 10.16% |
| Deliveries | 138 45.25% | 68 22.30% | 34 11.15% | 21 6.89% | 6 1.97% | 4 1.31% | 34 11.15% |
| Supply chains management | 153 50.16% | 50 16.39% | 40 13.11% | 18 5.90% | 9 2.95% | 3 0.98% | 32 10.49% |
| Scheduling problems | 128 41.97% | 52 17.05% | 52 17.05% | 28 9.18% | 6 1.97% | 6 1.97% | 33 10.82% |
| Manufacturing processes monitoring | 151 49.51% | 59 19.34% | 34 11.15% | 21 6.89% | 7 2.30% | 4 1.31% | 29 9.51% |
| Anomaly detection | 119 39.02% | 58 19.02% | 37 12.13% | 23 7.54% | 29 9.51% | 8 2.62% | 31 10.16% |
| Computer vision | 108 35.41% | 61 20.00% | 43 14.10% | 27 8.85% | 26 8.52% | 13 4.26% | 27 8.85% |
| Optimization | 77 25.25% | 63 20.66% | 45 14.75% | 41 13.44% | 32 10.49% | 15 4.92% | 32 10.49% |
| Cognitive systems | 136 44.59% | 63 20.66% | 34 11.15% | 25 8.20% | 10 3.28% | 3 0.98% | 34 11.15% |
| Autonomous systems | 117 38.36% | 64 20.98% | 43 14.10% | 31 10.16% | 18 5.90% | 4 1.31% | 28 9.18% |
| Robots | 124 40.66% | 60 19.67% | 36 11.80% | 34 11.15% | 17 5.57% | 6 1.97% | 28 9.18% |

In the next open-ended question, the students indicated other applications that they have used AI. This question was answered by 44 students who indicated at least one

application that was not covered in the previous question. The AI applications reported by students are presented in Table 15 (the values in parentheses indicate the number of students pointing to a given application if more than one).

**Table 15.** Additional applications of AI reported by students.

| AI Applications | | |
|---|---|---|
| Ad-click prediction | Finance (2) | Natural language processing |
| Agriculture (2), e.g., farm | Game development (4), e.g., | OCR systems |
| fields observation | tic-tac-toe | Process optimization and |
| Application development | Hair region segmentation | research |
| Automated data exploration | Image classification | Recommendation systems |
| Chatbots/Conversational | Inverse problems solving | (2) |
| agents (2) | IoT | Research on creativity |
| Complex networks | Healthcare/Medicine (7), e.g., | Risk stratification |
| Computer vision | Bioinformatics, Biomedical image | Scheduling problem with |
| Cross-silo federated learning | analysis, Calorie counter, Disease | quality aspects |
| Cybersecurity | prediction, Drug repurposing, | Sequence models |
| Digital culture | Medical data classification, | Solving mathematical |
| Digital marketing | Survival prediction | puzzles |
| Expert systems | Minimal tasks | Spam/ham classification |
| Finite element modeling | Network monitoring and security | Sports analytics |
| | | Telecom |
| | | Text mining |

Some students also identified other aspects of AI in this question without assigning them to a specific practical application: analytics, classification, forecasting, neural networks, TensorFlow, zero-shot learning.

The next open-ended question was about what kind of AI-projects students have implemented in the past. Ninety-eight students answered this question. Figure 5 shows the visualization of the frequency of AI-projects implemented by the students. The size of the rectangles is proportional to the frequency of responses. The largest number of projects concerned problems of classification and prediction or forecasting. Taking into account the application domains of the projects, the most frequently mentioned were medical diagnosis, chatbot development, object detection, autonomous vehicles development, and anomaly detection.

Detailed analysis of the survey data confirmed a strong relationship between the level of students' knowledge of AI and participation in the projects. In order to check this relationship, the AI knowledge index was used, calculated as follows for each student:

1.  We take into account 65 responses to close-ended questions covering only AI-related issues;
2.  We assign the following values to answer variants: 0 "not at all" and empty answer, 1 "to a small extent", 2 "to some extent", 3 "to a moderate extent", 4 "to a great extent", 5 "to a very great extent";
3.  We sum up the values from all 65 responses for a given student.

The questionnaires were divided into two groups: students participating in at least one AI project and students not participating in any AI project. Basic statistics on the AI knowledge index were calculated for both groups. The results are presented in Table 16. To visually compare the AI knowledge index in both groups, histograms (Figure 6) and box–whisker plots (Figure 7) were prepared. Both the plots and the calculated statistics show that, in the group of students involved in AI projects, higher values of the AI knowledge index analyzed prevail.

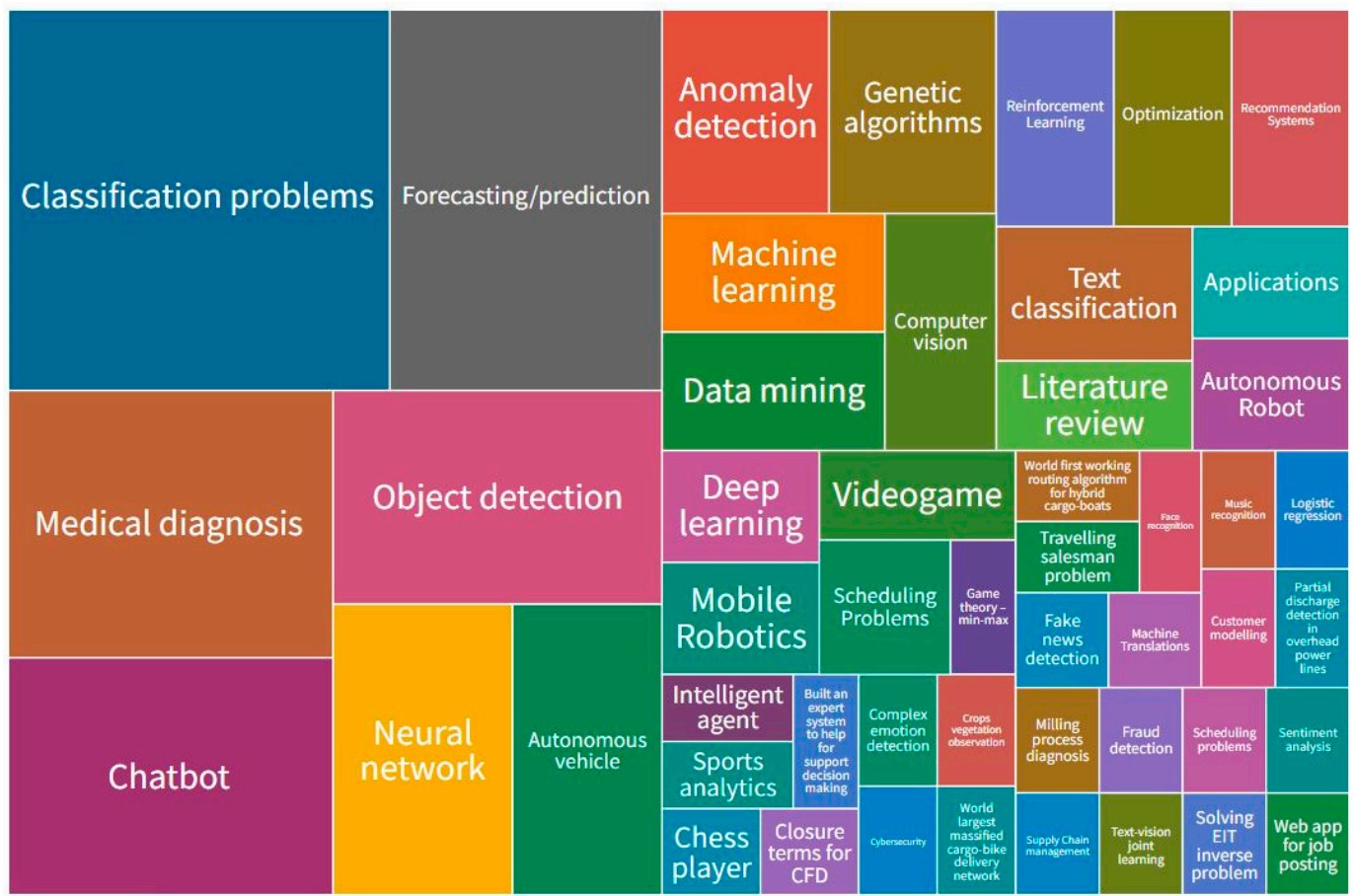

**Figure 5.** Topics of AI projects indicated by the students during the survey; the size of the boxes indicates the frequency of the answer.

**Table 16.** Statistics on the AI knowledge index.

| Group of Students | No. | Min | Mode | Median | Arithmetic Mean | Max |
|---|---|---|---|---|---|---|
| Not involved in any AI project | 207 | 0 | 26 | 59 | 68.74 | 264 |
| Involved in at least one AI project | 98 | 7 | 110 | 110 | 112.69 | 294 |

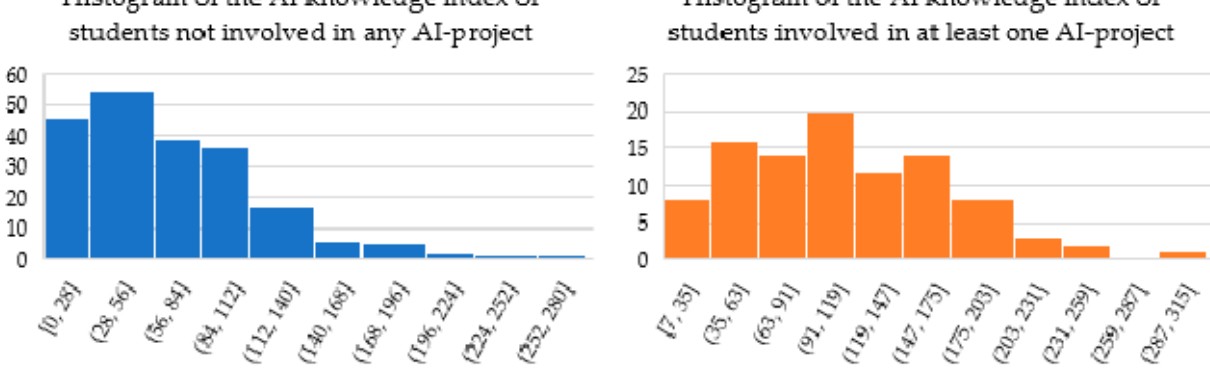

**Figure 6.** Histograms concerning the AI knowledge index for the two compared groups of students.

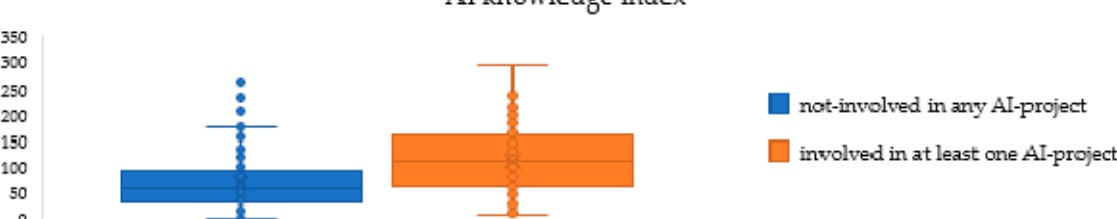

**Figure 7.** Box–whisker plots concerning the AI knowledge index for the two compared groups of students.

Additionally, the Mann–Whitney U test was used to show that the difference between the two groups is statistically significant (the significance level $\alpha = 0.05$). The following null hypothesis (H0) and the alternative hypothesis (H1) were formulated:

**Hypothesis 0 (H0).** *The two groups come from the same population*;

**Hypothesis 1 (H1).** *The two groups come from different populations*.

The results obtained for the Mann–Whitney U test are summarized in Table 17. The *p*-value turns out to be approximately 0. Therefore, for the assumed significance level, the null hypothesis should be rejected. This means that, with a 95% probability, it can be concluded that students participating in AI projects are characterized by a higher level of knowledge in the field of AI compared to other students. This shows that providing students with the opportunity to participate in projects can be crucial in developing their competences. It is particularly desirable when these projects relate to practical applications, such as in the case of the discussed group of respondents. The students who have engaged in projects receive a portion of practical skills that will surely bear fruit in the labor market when faced with the real challenges of the industry.

**Table 17.** Results obtained for the Mann–Whitney U test.

| Sum of Ranks for the "Project" Group | Sum of Ranks for the "Non-Project" Group | U Statistic | Z Statistic | *p*-Value |
|---|---|---|---|---|
| 19,579 | 27,086 | 5558 | 6.374 | 0.000 |

The last close-ended question in the AI section was about learning techniques such as lectures and labs. The students were asked to indicate to what extent these learning techniques are useful in teaching AI. Table 18 presents the students' answers to this question. According to the students, laboratory classes are the most useful. More than half of the respondents indicated a high or very high usefulness of this form of classes, following this are project-based learning (individual work), project-based learning (teamwork), and workshops. The least useful are lectures, e-learning, and general review of an issue. Generally speaking, each of the learning techniques analyzed has its supporters—at least 20% of the students indicated the high or very high usefulness of each technique.

There is a clear preference towards AI teaching practical skills in laboratory classes, projects, and workshops. This observation coincides with the conclusion formulated in the previous question that the practical realization of some tasks (e.g., in a project) gives students the most knowledge and skills in the field of AI.

In the next open-ended question, students could write what, in their opinion, is **the most difficult in learning AI**. One hundred and seventy respondents gave an answer that indicated at least one difficult issue. By far the largest number of students assessed that mastering math is the most difficult thing to learn AI. The difficulty in understanding the mathematics behind the AI algorithms was particularly emphasized (algebra, calculus, a lot of equations and formulas, statistics, probability, and mathematical logic).

**Table 18.** The students' assessment of the usefulness of the learning techniques to teach AI.

| Learning Technique | Not at All | To a Small Extent | To Some Extent | To a Moderate Extent | To a Great Extent | To a Very Great Extent | ND |
|---|---|---|---|---|---|---|---|
| Lectures | 24 7.87% | 31 10.16% | 77 25.25% | 64 20.98% | 55 18.03% | 32 10.49% | 22 7.21% |
| Labs | 7 2.30% | 18 5.90% | 37 12.13% | 55 18.03% | 70 22.95% | 92 30.16% | 26 8.52% |
| Workshops | 11 3.61% | 20 6.56% | 48 15.74% | 60 19.67% | 61 20.00% | 75 24.59% | 30 9.84% |
| Project-based learning (individual work) | 11 3.61% | 18 5.90% | 36 11.80% | 51 16.72% | 74 24.26% | 83 27.21% | 32 10.49% |
| Project-based learning (teamwork) | 13 4.26% | 19 6.23% | 34 11.15% | 44 14.43% | 86 28.20% | 79 25.90% | 30 9.84% |
| Problem-based Learning | 16 5.25% | 21 6.89% | 40 13.11% | 53 17.38% | 78 25.57% | 65 21.31% | 32 10.49% |
| E-learning | 23 7.54% | 43 14.10% | 75 24.59% | 64 20.98% | 39 12.79% | 31 10.16% | 30 9.84% |
| General review of an issue | 32 10.49% | 49 16.07% | 70 22.95% | 55 18.03% | 41 13.44% | 29 9.51% | 29 9.51% |

The second most frequently mentioned problem was understanding the theoretical issues related to AI. The students mentioned the following problems that made it difficult to understand the concept and intuition accompanying AI methods: the complex structure of AI issues, the "black-box nature" of many AI applications, many different approaches to solving problems, and the wide variety of AI areas (individual AI areas are very different from each other). Some people found it difficult to start learning AI, getting basic information about AI that would allow them to master the right techniques and software for AI. Mastering the basic issues was found to be hindered by an overloading of courses (too much information provided during one class), the lack of a source of good-quality books, educational and training materials, or a problem with obtaining them, and the lack of a good internship with a competent mentor. The lack of appropriate data sets and hardware for AI was also noted.

The abovementioned problems are also related to the third of the most frequently raised difficulties regarding AI—a relatively large number of students emphasized the difficulty in mastering complex algorithms and programming languages, the lack of a clear introduction to programming, especially in TensorFlow, Keras, PyTorch, and Numpy. Some people pointed out that it is difficult to transfer theoretical knowledge to applications in real problems (transferring theory to code). In their opinion, it is caused by the lack of a clear implementation strategy. Implementation issues are also complicated by expensive tools that allow AI to be applied in practice.

Some respondents mentioned the most difficult applications and areas of AI: computer vision, natural language processing, reinforcement learning, deep learning, and ML. The students also listed some AI-related tasks that are particularly demanding: designing a neural network model (finding suitable model, optimization of network architecture for a given data set), understanding how neural networks works, selecting the most optimal method for a given problem or data, understanding which parameters are important and why, development of a generative adversarial networks, generalization, evaluation, feature extraction, implementation of AI agents, and debugging.

Several respondents highlighted the data and the problems that could be associated with it: data management, data understanding, data preprocessing, data analysis, the quality of data under analysis, and data transformation in such a way that a computer can interpret the data.

Some students also pointed to general problems that make studying difficult, such as too little free time (respondents emphasized the need to spend a lot of time studying

the basics of AI and programming languages) or staying motivated and focused. It is also associated with the need to keep up with recent approaches and developments in the field of AI.

To sum up, the students most often mentioned difficulties with:

- Understanding the concept of AI issues and how AI works—the complexity of issues causes a barrier that is difficult to overcome for people who want to start learning AI;
- Mathematical issues behind AI;
- High complexity of algorithms;
- Programming languages and coding;
- Finding the right resources for tutorials and other materials to study.

In the next open-ended question, students were asked **what is missing in the AI education process**. One hundred and forty-three students answered this question by giving at least one missing factor. The most common answer was the lack of practical elements in the AI education process, therefore an equal balance between theory and practice is missing. The students suggested that there should be more practical applications of AI (teaching AI techniques by applying them on a project that can be chosen), as well as more real-life examples, real-world datasets, activities, workshops, and solving of simple tasks. According to the respondents, showing the practical context of AI would facilitate the implementation of AI solutions. There is also no information on how to integrate different AI platforms, train AI models on cloud platforms, deploy models, and perform online learning. It is also desirable to create a centralized place to receive proven tested techniques, an environment to experiment on projects, and a well-organized online site with projects to work on for self-study and in groups, with a forum where people can help each other.

According to the students, the number of laboratory and project classes in universities should be increased. This remark was also one of the most frequent. Classes should be rich in a variety of case studies and education could be carried out in partnership with some companies, which is currently missing. There is also no easy access to sophisticated AI software, modern tools, easy-to-understand articles, books, and other materials to deepen knowledge (especially in native languages).

In addition to the emphasis on practical education, some students also suggested that attention should be paid to theoretical issues that will enable an understanding of the necessary basics (e.g., how an AI works, how AI is coded, what can AI do with data, and why a specific approach works).

The answers also include proposals for some modifications to the course of AI education: showing the basic things of AI in earlier years of education (high school), starting AI teaching from bachelor level, introducing performance-based learning, more programming classes, more lessons about neural networks, using open source tools such as TensorFlow, using online open classrooms, teaching AI related to society, teaching how to overcome the difficulties of developing AI nowadays due to ethical reasons. Apart from that, the students suggested increasing the emphasis on the importance of data, understanding algorithms, and mathematical background (a better explanation of the fundamental mathematics and how they relate to the algorithm implementations). Other important postulations were that the education process should be more specialized in specific fields of AI, that exercises should be based on real problems which students have to face in everyday life, and that the content of education should be updated to keep up with any novelties related to AI (e.g., modern AI applications).

Some respondents pointed to the general shortcomings of the education process, such as: too few hours for learning AI, too few competent teachers, poor access to simple explanations of AI, and good classes.

Summing up, the students mainly point out the following shortcomings:

- Too few practical issues and applications of AI in education;
- Lack of real-life case studies drawn primarily from business and companies;
- Too few laboratories, projects, and workshops in AI education;

- Insufficient mathematical preparation of students to learn AI;
- Poor access to good-quality study materials and modern AI tools, which will be constantly updated in line with emerging novelties;
- Too little emphasis on a good understanding of the basics of AI during education.

The last question in the AI section was about **things that could support students in the AI learning process**. One hundred and twenty-eight respondents answered this question by giving at least one thing that could support them in learning AI. The observations after analyzing these answers are similar to the conclusions from the previous question. Many students pay attention to those aspects that have already been raised in the question of the shortcomings in AI education.

The students most often mentioned the need to put more emphasis on practical examples of AI application, increasing the number of project classes, workshops, and labs, learning by doing, teamwork, additional classes for non-advanced students (teaching step by step with a large number of examples and case studies), and teaching programming (especially in Python). Several respondents pointed to the usefulness of internships, learning with individual assistance, and indicated specific solutions to support learning (IBM Labs, Coursera, virtual machines with implemented CUDA graphic cards).

Some students need more material and tutorials available online, including the state of the art about techniques in every domain of AI as well as books that do not only focus on the AI structure, but also on problems AI can solve. The students suggested that materials and exercises to study AI could be collected in the form of a consistent platform.

Also important are access to open source software, access to large quantities of computing resources, a centralized place to receive proven and tested techniques, and infrastructure (e.g., robots) on which AI applications can be tested. There is also a proposal to organize competitions to solve some problems with AI (such as on the Kaggle platform).

The students also paid attention to general learning needs: access to qualified teachers and to professionals (interaction with companies), financial support, funded projects that can provide students with the necessary equipment in AI learning process, structured learning paths with theoretical references for a deeper AI understanding, summer schools, and access to more free time.

Summing up, apart from the needs mentioned in the previous question, the students in this question pointed to the following needs:

- Participation in internships, joint international projects, and competitions;
- Infrastructure for testing AI applications;
- Appropriate equipment (more computing resources);
- Contact with companies and professionals;
- Online base of educational materials and AI tools and more programming lessons.

*7.3. Internet of Things*

This section presents the level of students' knowledge connected with IoT. The percentages calculated in the tables below (from Tables 19–23) assume that 100% is the number of students who chose "yes" in the question "Have you ever learned about IoT?".

Table 19 summarizes the declared students' knowledge in the field of 22 selected IoT topics. It can be seen that almost all students who have studied IoT know the background information about IoT. About 95% of the students also have at least a basic knowledge of IoT application scenarios. Looking at examples of practical issues related to IoT, students' knowledge is more diverse. The best-known topics are computer networking, IoT architecture, and sensors. On the opposite side, there are topics such as: machine-to-machine (M2M) industrial IoT protocols, searching for vulnerabilities, distribution of computing processes in IoT networks, IoT maintenance, and cryptography. In each of these topics, at least 30% of the students did not have any knowledge at all. Focusing on these weakly known topics may be one of the most important needs in IoT education.

**Table 19.** The students' knowledge connected with the selected IoT topics.

| IoT Topic | Not at All | To a Small Extent | To Some Extent | To a Moderate Extent | To a Great Extent | To a Very Great Extent | ND |
|---|---|---|---|---|---|---|---|
| General information about IoT | 1 0.72% | 19 13.67% | 29 20.86% | 53 38.13% | 23 16.55% | 7 5.04% | 7 5.04% |
| Application scenarios of IoT | 7 5.04% | 18 12.95% | 32 23.02% | 39 28.06% | 23 16.55% | 6 4.32% | 14 10.07% |
| IoT Architecture | 15 10.79% | 30 21.58% | 40 28.78% | 27 19.42% | 10 7.19% | 5 3.60% | 12 8.63% |
| IoT deployment | 27 19.42% | 29 20.86% | 35 25.18% | 21 15.11% | 10 7.19% | 4 2.88% | 13 9.35% |
| IoT components | 20 14.39% | 31 22.30% | 33 23.74% | 24 17.27% | 16 11.51% | 4 2.88% | 11 7.91% |
| M2M industrial IoT protocols | 54 38.85% | 34 24.46% | 18 12.95% | 11 7.91% | 7 5.04% | 2 1.44% | 13 9.35% |
| Sensors | 17 12.23% | 26 18.71% | 28 20.14% | 29 20.86% | 19 13.67% | 8 5.76% | 12 8.63% |
| IoT devices programming | 29 20.86% | 29 20.86% | 25 17.99% | 21 15.11% | 14 10.07% | 8 5.76% | 13 9.35% |
| IoT maintenance | 43 30.94% | 40 28.78% | 23 16.55% | 13 9.35% | 4 2.88% | 2 1.44% | 14 10.07% |
| Distribution of computing processes in IoT nets | 44 31.65% | 37 26.62% | 24 17.27% | 13 9.35% | 5 3.60% | 2 1.44% | 14 10.07% |
| Computer Networking | 11 7.91% | 32 23.02% | 36 25.90% | 31 22.30% | 15 10.79% | 6 4.32% | 8 5.76% |
| Data analytics | 23 16.55% | 28 20.14% | 33 23.74% | 28 20.14% | 10 7.19% | 6 4.32% | 11 7.91% |
| Cloud computing | 21 15.11% | 37 26.62% | 30 21.58% | 24 17.27% | 9 6.47% | 5 3.60% | 13 9.35% |
| Databases development | 21 15.11% | 28 20.14% | 30 21.58% | 30 21.58% | 11 7.91% | 6 4.32% | 13 9.35% |
| Data transfer protocols | 21 15.11% | 25 17.99% | 32 23.02% | 31 22.30% | 11 7.91% | 6 4.32% | 13 9.35% |
| IoT Communication Terminals and Gateways | 33 23.74% | 33 23.74% | 33 23.74% | 13 9.35% | 12 8.63% | 2 1.44% | 13 9.35% |
| Knowledge management | 32 23.02% | 41 29.50% | 26 18.71% | 19 13.67% | 6 4.32% | 1 0.72% | 14 10.07% |
| Cybersecurity | 33 23.74% | 35 25.18% | 20 14.39% | 27 19.42% | 8 5.76% | 5 3.60% | 11 7.91% |
| Cryptography | 42 30.22% | 25 17.99% | 17 12.23% | 28 20.14% | 9 6.47% | 5 3.60% | 13 9.35% |
| Basic Network Attacks | 39 28.06% | 22 15.83% | 27 19.42% | 21 15.11% | 13 9.35% | 4 2.88% | 13 9.35% |
| Real Time Operating Systems | 31 22.30% | 30 21.58% | 27 19.42% | 22 15.83% | 10 7.19% | 6 4.32% | 13 9.35% |
| Searching for Vulnerabilities | 51 36.69% | 31 22.30% | 18 12.95% | 13 9.35% | 9 6.47% | 2 1.44% | 15 10.79% |

The next question checked whether students know how to apply IoT in different contexts. Table 20 lists the nine contexts of using IoT. It can be seen that students' knowledge in all of the contexts considered is lower compared to the general knowledge of IoT from the previous question. A high or very high level of knowledge in all contexts was declared by less than 12% of the students. In all contexts, at least 23% of the students declared their lack of any knowledge of using IoT. It can be concluded that all the considered contexts of IoT require more attention in the education process, particularly market behavior, logistics, and deliveries, which are characterized by the lowest level of students' knowledge.

**Table 20.** The students' knowledge about how to use IoT in different context.

| Context of Using IoT | Not at All | To a Small Extent | To Some Extent | To a Moderate Extent | To a Great Extent | To a Very Great Extent | ND |
|---|---|---|---|---|---|---|---|
| Quality problems | 42 30.22% | 28 20.14% | 25 17.99% | 21 15.11% | 6 4.32% | 4 2.88% | 13 9.35% |
| Machine condition monitoring | 39 28.06% | 32 23.02% | 23 16.55% | 17 12.23% | 8 5.76% | 6 4.32% | 14 10.07% |
| Robotics | 32 23.02% | 35 25.18% | 27 19.42% | 16 11.51% | 10 7.19% | 6 4.32% | 13 9.35% |
| Deliveries | 53 38.13% | 34 24.46% | 17 12.23% | 11 7.91% | 3 2.16% | 4 2.88% | 17 12.23% |
| Market behavior | 57 41.01% | 34 24.46% | 14 10.07% | 13 9.35% | 3 2.16% | 2 1.44% | 16 11.51% |
| Data management | 33 23.74% | 34 24.46% | 29 20.86% | 15 10.79% | 11 7.91% | 3 2.16% | 14 10.07% |
| Support decision-making | 44 31.65% | 34 24.46% | 18 12.95% | 15 10.79% | 10 7.19% | 3 2.16% | 15 10.79% |
| Process parameters monitoring | 43 30.94% | 36 25.90% | 22 15.83% | 13 9.35% | 6 4.32% | 5 3.60% | 14 10.07% |
| Logistics | 54 38.85% | 34 24.46% | 18 12.95% | 12 8.63% | 3 2.16% | 4 2.88% | 14 10.07% |

In the next question, the students were asked to indicate the usefulness of 11 selected techniques in the context of IoT learning. Table 21 presents students' answers to this question. Table 21 shows that, according to students, the application programming interface (API) is most helpful in mastering IoT. More than 25% indicated the great usefulness of API, and, for more than 13%, the usefulness of API is very high. One of the highest results is also recorded for data processing and transformation, as well as big data management. The usefulness of data processing and management techniques is not surprising, due to the growing importance of these issues for modern business and industry. The lowest usefulness in learning IoT has digital twin, infrastructure as a service, platform as a service, and software as a service.

In the IoT section of the survey, the students were also asked about the usefulness of different forms of education in teaching IoT. The results (Table 22) are analogous to the results obtained in the AI section. It can be seen that, according to the students, the most useful are project-based learning, laboratory classes, and workshops. The least important are e-learning, general review of an issue, and lectures. However, for a large part of students, these three forms of education also have some useful features, especially the general review of an issue, for which high or very high usefulness was indicated by about 30% of the students. Therefore, the results of this question cannot be the reason for resigning from the three mentioned forms of education.

The last close-ended question in the IoT section of the survey concerned the software/environment used in IoT. The students indicated to what extent they use selected software and environments. The students' responses are summarized in Table 23. It can be seen that the most used tool is Arduino IoT. About 63% of the students answered that they use this tool to at least a small extent. In turn, the remaining four tools are used to a much lower extent. More than half of the students do not use Amazon Web Services (AWS) Lambda, and MapReduce at all.

The next question was an open-ended question. The students could indicate a different software/environment, not mentioned in the previous question, that they use in the context of IoT. Twenty-three students answered by pointing to at least one IoT tool. The tools they indicate are as follows (the values in parentheses indicate the number of students pointing to a given tool): Raspberry Pi (4) (respondents' answers: Raspberry, Raspberry Pi, Raspbian), Python (2), Blynk, Codesys, Druid, freeRTOS, Eclipse, ESP IDF, Flink, Heidi SQL, IntelliJ dev tool with modbus simulator, Kafka, Linux (Ubuntu), Mosquitto MQTT,

MTConnect interface and internal software for data analysis, TMAC, BI, Node-RED, Oracle IoT, PyCharm, Spark, Thonny on Raspberry Pi 4, TinyOs, Visual Studio Code (2), Watson from IBM, WeBots for modeling. Four students pointed to the Arduino that appears in the previous question.

**Table 21.** The students' assessment of the usefulness of the following techniques to learn IoT.

| Context of Using IoT | Not at All | To a Small Extent | To Some Extent | To a Moderate Extent | To a Great Extent | To a Very Great Extent | ND |
|---|---|---|---|---|---|---|---|
| Digital twins | 24 17.27% | 17 12.23% | 32 23.02% | 29 20.86% | 6 4.32% | 10 7.19% | 21 15.11% |
| Big data management | 10 7.19% | 14 10.07% | 31 22.30% | 27 19.42% | 19 13.67% | 18 12.95% | 20 14.39% |
| Data processing and transformation | 8 5.76% | 12 8.63% | 27 19.42% | 29 20.86% | 25 17.99% | 17 12.23% | 21 15.11% |
| Data display | 11 7.91% | 15 10.79% | 32 23.02% | 34 24.46% | 13 9.35% | 11 7.91% | 23 16.55% |
| Industrial Automations | 12 8.63% | 10 7.19% | 30 21.58% | 24 17.27% | 22 15.83% | 17 12.23% | 24 17.27% |
| Anomaly detection | 16 11.51% | 8 5.76% | 26 18.71% | 24 17.27% | 28 20.14% | 15 10.79% | 22 15.83% |
| IaaS (Infrastructure as a Service) | 19 13.67% | 15 10.79% | 21 15.11% | 34 24.46% | 18 12.95% | 8 5.76% | 24 17.27% |
| PaaS (Platform as a Service) | 20 14.39% | 16 11.51% | 29 20.86% | 24 17.27% | 19 13.67% | 8 5.76% | 23 16.55% |
| SaaS (Software as a Service) | 17 12.23% | 15 10.79% | 30 21.58% | 18 12.95% | 29 20.86% | 8 5.76% | 22 15.83% |
| Containers and orchestrators | 15 10.79% | 13 9.35% | 24 17.27% | 36 25.90% | 19 13.67% | 10 7.19% | 22 15.83% |
| Application Programming Interface (API) | 8 5.76% | 14 10.07% | 17 12.23% | 25 17.99% | 35 25.18% | 19 13.67% | 21 15.11% |

**Table 22.** The students' assessment of the usefulness of the learning techniques to teach IoT.

| Learning Technique to Teach IoT | Not at All | To a Small Extent | To Some Extent | To a Moderate Extent | To a Great Extent | To a Very Great Extent | ND |
|---|---|---|---|---|---|---|---|
| Lectures | 6 4.32% | 18 12.95% | 31 22.30% | 32 23.02% | 23 16.55% | 16 11.51% | 13 9.35% |
| Labs | 2 1.44% | 4 2.88% | 18 12.95% | 31 22.30% | 33 23.74% | 36 25.90% | 15 10.79% |
| Workshops | 4 2.88% | 7 5.04% | 18 12.95% | 29 20.86% | 33 23.74% | 34 24.46% | 14 10.07% |
| Project-based learning (individual work) | 1 0.72% | 6 4.32% | 11 7.91% | 25 17.99% | 38 27.34% | 42 30.22% | 16 11.51% |
| Project-based learning (teamwork) | 2 1.44% | 3 2.16% | 14 10.07% | 24 17.27% | 37 26.62% | 45 32.37% | 14 10.07% |
| Problem-based learning | 5 3.60% | 3 2.16% | 14 10.07% | 29 20.86% | 39 28.06% | 34 24.46% | 15 10.79% |
| E-learning | 12 8.63% | 17 12.23% | 32 23.02% | 31 22.30% | 16 11.51% | 18 12.95% | 13 9.35% |
| General review of an issue | 10 7.19% | 10 7.19% | 30 21.58% | 33 23.74% | 25 17.99% | 17 12.23% | 14 10.07% |

**Table 23.** The level of use of the software/environment selected in IoT by students.

| Software/Environment | Not at All | To a Small Extent | To Some Extent | To a Moderate Extent | To a Great Extent | To a Very Great Extent | ND |
|---|---|---|---|---|---|---|---|
| MapReduce | 81<br>58.27% | 12<br>8.63% | 9<br>6.47% | 10<br>7.19% | 6<br>4.32% | 3<br>2.16% | 18<br>12.95% |
| Cloud Services & Serverless Technologies (AWS, GCP, DigitalOcean, Linode) | 60<br>43.17% | 19<br>13.67% | 13<br>9.35% | 14<br>10.07% | 9<br>6.47% | 5<br>3.60% | 19<br>13.67% |
| AWS Lambda | 88<br>63.31% | 13<br>9.35% | 5<br>3.60% | 12<br>8.63% | 3<br>2.16% | 0<br>0.00% | 18<br>12.95% |
| Azure functions | 71<br>51.08% | 12<br>8.63% | 13<br>9.35% | 16<br>11.51% | 8<br>5.76% | 1<br>0.72% | 18<br>12.95% |
| Arduino IoT | 36<br>25.90% | 24<br>17.27% | 19<br>13.67% | 21<br>15.11% | 15<br>10.79% | 9<br>6.47% | 15<br>10.79% |

The next open-ended question was about what kind of IoT projects students have implemented in the past. Thirty-nine students answered this question. Figure 8 shows the visualization of the frequency of IoT projects implemented by the students. The students mainly mentioned projects related to the use of IoT in home automation, robot mobility, agriculture automation, and healthcare. The projects were mainly related to skills such as monitoring, detection, radio communication, security, and the use of IoT microcomputers.

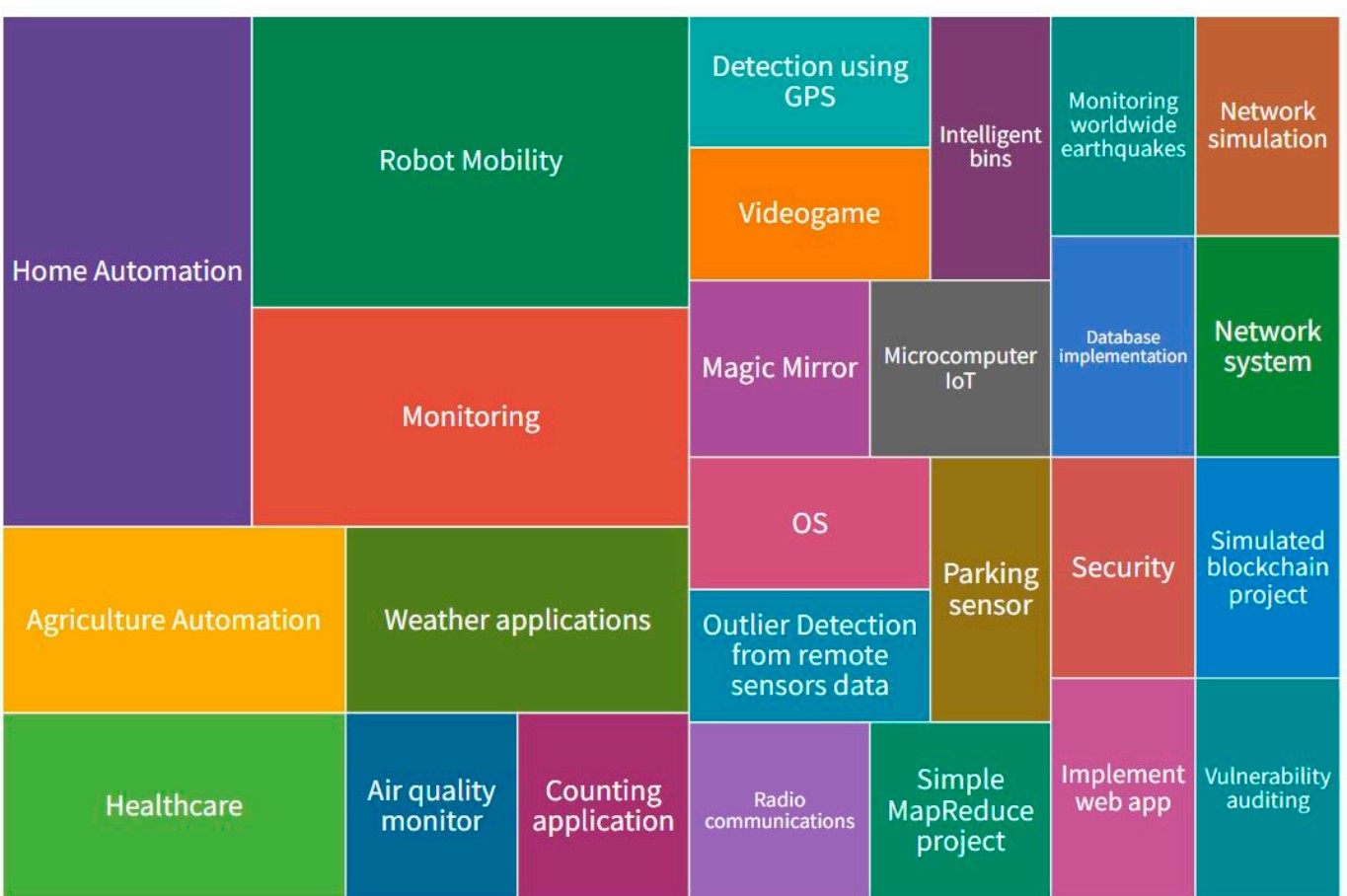

**Figure 8.** Topics of IoT projects indicated by the students during the survey; the size of the boxes indicates the frequency of the answer.

In the next question, the students wrote what is **the most difficult in learning IoT**. Thirty-seven respondents gave an answer that indicated at least one difficult issue. The

students most often emphasized the difficulties in understanding concepts related to IoT. This problem is mainly due to the high complexity of IoT. The high complexity requires acquiring a lot of information and knowledge to get started with IoT. It is also problematic to understand when something in IoT should be used and how to choose between different solutions.

The multitude of IoT applications became a problem for the students. IoT is a very broad term, and many things can be classified under it. Therefore, nailing down the concept is more difficult when there are so many examples and possibilities in IoT.

Many of the responses concerned the IoT infrastructure. Infrastructure issues were often mentioned as issues related to the general concept of IoT. In the context of infrastructure, the students indicated difficulties related to electronics (especially microcontrollers), protocols, sensors (especially connecting sensors to the Internet), the need to purchase an appropriate hardware, maintenance of IoT, combination of hardware and software interoperation, integration of multiple technologies and platforms, and finding an appropriate environment. Two students also noted difficulties in the area of cybersecurity and programming.

Several responses concerning issues that hinder the education process itself included a lack of adequate resources and correct documentation, the poor quality of code found on the Internet, and difficulty in applying concepts in a real environment to see how it could work.

In the next open-ended question, the students were asked **what is missing in the IoT education process**. Thirty-five students answered this question by giving at least one missing factor. The most common answer is the lack of an adequate number of project-based learning and laboratory classes, as well as too little emphasis on teaching practical skills related to IoT (too few solving real-world problems, too little variety of case studies, and the lack of application-driven education). The showing of complex designs from scratch is also a lack.

The students drew attention to the shortcomings in educating the basics of IoT, which makes it difficult to understand what IoT is and what its essence is. There is also a lack of practice on hardware, therefore students do not have the necessary experience to deal with IoT.

Some respondents indicated specific issues that receive insufficient attention in the education process. These are: privacy, operating systems, and practice on sensors. IoT learning is also hampered by the low availability of IoT in the content of courses at universities, as well as insufficient access to the standard documentation.

The last question in the IoT section was as follows: "What would be useful for you to **facilitate the learning process of IoT**?". Thirty-six respondents answered this question by giving at least one thing that could support them in IoT learning. The following needs arose from the students' responses:

- Teaching the theory behind IoT to understand when and why IoT is useful;
- More project-based learning and workshops (e.g., small weekly projects or projects combining all IoT techniques) including examples and practical activities (real scenarios and cases);
- More information on sensors, energy consumption, circuitry;
- Increasing the availability of IoT devices (digital twins, simulators) and easy-to-use technologies;
- More accessible information, materials, and online courses;
- Teaching C++, Python, JavaScript;
- Introducing subjects at universities entirely dedicated to IoT.

### 7.4. Edge Computing

This section contains information on the level of students' knowledge connected in the area of EC. The percentages calculated in the tables below (from Tables 24–29) assume that

100% is the number of students who chose "yes" in the question "Have you ever learned about EC?".

Table 24 shows students' knowledge in the field of five selected topics of EC. It can be seen that all students who have ever studied EC knew the general concept about it (in this question, three respondents did not answer). A similar situation is observed in the case of EC applications—all respondents who answered this question had at least a basic knowledge of EC applications. Looking at the other three EC topics, it can be seen that students' knowledge of them is slightly lower. None of the respondents assessed their knowledge at a very high level. In addition, about 30% of the students did not know these topics at all or knew them little. Thus, these topics (privacy and security, scalability and reliability, speed and efficiency) may indicate potential needs in the field of EC education.

**Table 24.** The students' knowledge about the selected topics in the area of EC.

| EC Topic | Not at All | To a Small Extent | To Some Extent | To a Moderate Extent | To a Great Extent | To a Very Great Extent | ND |
|---|---|---|---|---|---|---|---|
| General concept | 0 0.00% | 6 26.09% | 4 17.39% | 7 30.43% | 2 8.70% | 1 4.35% | 3 13.04% |
| Privacy and security | 3 13.04% | 4 17.39% | 6 26.09% | 4 17.39% | 2 8.70% | 0 0.00% | 4 17.39% |
| Scalability and reliability | 2 8.70% | 7 30.43% | 4 17.39% | 3 13.04% | 3 13.04% | 0 0.00% | 4 17.39% |
| Speed and efficiency | 1 4.35% | 6 26.09% | 6 26.09% | 4 17.39% | 2 8.70% | 0 0.00% | 4 17.39% |
| Applications | 0 0.00% | 7 30.43% | 3 13.04% | 6 26.09% | 3 13.04% | 0 0.00% | 4 17.39% |

In the next question, the students assessed to what extent they know six selected technologies that are used in EC implementations. The data obtained are recorded in Table 25. The data show that the smallest percentage of the students knew the Azure edge (47.83%). Additionally, it can be seen that, in the case of three technologies (mobile EC, fog computing, service composition, and service-oriented computing), none of the students declared a high or very high level of knowledge. Thus, education on these topics may be particularly needed to supplement students' knowledge.

**Table 25.** The students' knowledge about the following technologies used in EC implementation.

| Technology in EC Implementation | Not at All | To a Small Extent | To Some Extent | To a Moderate Extent | To a Great Extent | To a Very Great Extent | ND |
|---|---|---|---|---|---|---|---|
| Mobile Edge Computing | 4 17.39% | 7 30.43% | 1 4.35% | 8 34.78% | 0 0.00% | 0 0.00% | 3 13.04% |
| Fog computing | 3 13.04% | 6 26.09% | 4 17.39% | 5 21.74% | 0 0.00% | 0 0.00% | 5 21.74% |
| Service composition and service-oriented computing | 5 21.74% | 5 21.74% | 2 8.70% | 7 30.43% | 0 0.00% | 0 0.00% | 4 17.39% |
| Micro data centers | 3 13.04% | 8 34.78% | 6 26.09% | 1 4.35% | 1 4.35% | 0 0.00% | 4 17.39% |
| Container technology | 2 8.70% | 7 30.43% | 3 13.04% | 4 17.39% | 2 8.70% | 0 0.00% | 5 21.74% |
| Azure edge | 7 30.43% | 5 21.74% | 1 4.35% | 4 17.39% | 1 4.35% | 0 0.00% | 5 21.74% |

The next question was an open-ended question in which the students could indicate what other technologies used in EC implementation they know. Only one student responded to this question by indicating a different technology than those mentioned in the previous question. The student indicated that he/she also knew the technology called distributed systems.

In addition to the technologies used in the EC implementation, the students were also asked about their knowledge of algorithms and techniques related to the EC implementation. The results obtained are presented in Table 26. The algorithms/techniques related to effective data collection, aggregation, and transportation were known for the highest percentage of students (69.57%). It can be seen that the same percentage of students was familiar with distributed computing and energy efficiency at least to a small extent. The lowest number of respondents declared knowledge about containerization (56.52%). However, the differences in the results obtained for the issues discussed here are small; therefore, it is difficult to indicate a clearly outlying issue in the context of students' knowledge.

**Table 26.** The students' knowledge on the following algorithms/techniques used in EC implementation.

| Algorithm/Technique in EC Implementation | Not at All | To a Small Extent | To Some Extent | To a Moderate Extent | To a Great Extent | To a Very Great Extent | ND |
|---|---|---|---|---|---|---|---|
| Distributed computing | 3 13.04% | 7 30.43% | 4 17.39% | 2 8.70% | 2 8.70% | 1 4.35% | 4 17.39% |
| Distributed storage | 5 21.74% | 6 26.09% | 4 17.39% | 3 13.04% | 0 0.00% | 1 4.35% | 4 17.39% |
| Reliability and fault tolerance | 5 21.74% | 2 8.70% | 3 13.04% | 7 30.43% | 1 4.35% | 0 0.00% | 5 21.74% |
| Containerization | 6 26.09% | 3 13.04% | 6 26.09% | 3 13.04% | 1 4.35% | 0 0.00% | 4 17.39% |
| Energy efficiency | 3 13.04% | 7 30.43% | 5 21.74% | 3 13.04% | 1 4.35% | 0 0.00% | 4 17.39% |
| Data replication | 5 21.74% | 3 13.04% | 4 17.39% | 5 21.74% | 2 8.70% | 0 0.00% | 4 17.39% |
| Efficiently collecting, aggregating, and moving data | 3 13.04% | 5 21.74% | 6 26.09% | 4 17.39% | 1 4.35% | 0 0.00% | 4 17.39% |

In the next question, the students assessed the extent to which they carry out activities related to EC. They were asked about eight activities. The results are presented in Table 27. The three activities mentioned least frequently in this question may indicate deficiencies in the educational process. It can be seen that they are: designing an edge computing architecture, implementing software solutions using EC middlewares, as well as doing data analytics in EC environments. These activities can be identified as one of the most essential needs that should be emphasized in the EC education process.

**Table 27.** The students' assessment of EC-connected activities.

| To What Extent Do You … | Not at All | To a Small Extent | To Some Extent | To a Moderate Extent | To a Great Extent | To a Very Great Extent | ND |
|---|---|---|---|---|---|---|---|
| identify the challenges of EC? | 3 13.04% | 5 21.74% | 6 26.09% | 3 13.04% | 1 4.35% | 0 0.00% | 5 21.74% |
| design an EC architecture? | 6 26.09% | 4 17.39% | 5 21.74% | 2 8.70% | 1 4.35% | 0 0.00% | 5 21.74% |
| describe the differences between edge, fog, cloud, and pervasive computing? | 3 13.04% | 7 30.43% | 5 21.74% | 2 8.70% | 1 4.35% | 0 0.00% | 5 21.74% |
| implement software solutions using EC middlewares? | 6 26.09% | 5 21.74% | 4 17.39% | 2 8.70% | 1 4.35% | 0 0.00% | 5 21.74% |
| understand the strengths and weaknesses of an EC architecture? | 3 13.04% | 7 30.43% | 4 17.39% | 3 13.04% | 1 4.35% | 0 0.00% | 5 21.74% |
| develop an edge computing project? | 5 21.74% | 8 34.78% | 1 4.35% | 3 13.04% | 1 4.35% | 0 0.00% | 5 21.74% |
| read papers related to EC? | 4 17.39% | 6 26.09% | 1 4.35% | 4 17.39% | 2 8.70% | 1 4.35% | 5 21.74% |
| do data analytics in EC environments? | 6 26.09% | 9 39.13% | 0 0.00% | 2 8.70% | 1 4.35% | 0 0.00% | 5 21.74% |

The next question was to what extent the students use hardware/software that enables them to use the EC platforms. Table 28 summarizes the responses provided. It can be seen that none of the five tools in this question are particularly widespread among students. The highest number of respondents indicated that they use Azure IoT Edge to a small extent (30.43%). However, at least 47.83% of the students have never used this tool. In the case of other tools, the percentage of the students who have never used them is even greater. Therefore, education in all of the hardware and software mentioned in this question is a potential need to broaden the skills involved in using these tools that enable EC platforms.

**Table 28.** The use of hardware/software that enables EC platforms by students.

| Hardware/Software | Not at All | To a Small Extent | To Some Extent | To a Moderate Extent | To a Great Extent | To a Very Great Extent | ND |
|---|---|---|---|---|---|---|---|
| FPGAs | 12 52.17% | 5 21.74% | 1 4.35% | 0 0.00% | 1 4.35% | 0 0.00% | 4 17.39% |
| Edge accelerators | 13 56.52% | 5 21.74% | 0 0.00% | 0 0.00% | 1 4.35% | 0 0.00% | 4 17.39% |
| Azure IoT Edge | 11 47.83% | 7 30.43% | 0 0.00% | 0 0.00% | 1 4.35% | 0 0.00% | 4 17.39% |
| AWS IoT Greengrass | 15 65.22% | 2 8.70% | 1 4.35% | 0 0.00% | 1 4.35% | 0 0.00% | 4 17.39% |
| RTOS | 15 65.22% | 2 8.70% | 1 4.35% | 0 0.00% | 1 4.35% | 0 0.00% | 4 17.39% |

In the next question, the students could indicate another software/environment they use in an EC context. Only one of the students answered this question by giving the following answer: "We made our own for a university project".

The last close-ended question in the survey concerned the extent to which students knew the possibilities of using EC in nine selected areas. Table 29 shows the level of students' knowledge on EC in these areas. It can be seen that none of the students declared knowledge of the highest level in any of the EC applications mentioned. The highest percentage of the students had knowledge of EC in the context of I4.0, but 43.48% of students had only a low level of knowledge in this area. The fewest students indicated that they had any knowledge of EC applications such as autonomous products, an autonomous production planning system, augmented reality, and autonomy in energy networks. Increasing the level of knowledge in these EC applications may be the most desirable due to, among other things, the importance of these applications in today's enterprises that implement the I4.0 concept.

The next open-ended question was about what kind of EC projects students have implemented in the past. Two students answered this question. Students listed the following projects related to EC areas:

- Data replication algorithms, exclusion algorithms, fault tolerance algorithms;
- Data replication, shared memory vs. shared nothing, distributed exclusion algorithm (Ricart–Agrawala and Lamport).

In the next question, the students wrote what is **the most difficult in learning EC**. Two respondents answered this question. According to the students, the most difficult aspect in learning EC are application examples and the implementation of some algorithms.

Moreover, the students were asked **what is missing in the EC education process**, but the students did not present their opinion on this topic.

The last question in the EC section was as follows: "What would be useful for you to **facilitate the learning process of EC**?". Two students answered this question as follows:

- Learning about it a little bit sooner;
- Online seminars.

**Table 29.** The level of students' knowledge on EC in the following applications.

| EC Application | Not at All | To a Small Extent | To Some Extent | To a Moderate Extent | To a Great Extent | To a Very Great Extent | ND |
|---|---|---|---|---|---|---|---|
| Autonomous machines | 6 26.09% | 8 34.78% | 3 13.04% | 0 0.00% | 2 8.70% | 0 0.00% | 4 17.39% |
| Autonomous production planning system | 8 34.78% | 8 34.78% | 1 4.35% | 2 8.70% | 0 0.00% | 0 0.00% | 4 17.39% |
| Augmented reality | 8 34.78% | 7 30.43% | 3 13.04% | 1 4.35% | 0 0.00% | 0 0.00% | 4 17.39% |
| Mobile agents (e.g., drones) | 5 21.74% | 8 34.78% | 4 17.39% | 1 4.35% | 1 4.35% | 0 0.00% | 4 17.39% |
| Autonomous products | 9 39.13% | 6 26.09% | 3 13.04% | 1 4.35% | 0 0.00% | 0 0.00% | 4 17.39% |
| Autonomy in energy networks | 8 34.78% | 7 30.43% | 1 4.35% | 2 8.70% | 1 4.35% | 0 0.00% | 4 17.39% |
| Facial recognition algorithms | 6 26.09% | 5 21.74% | 4 17.39% | 2 8.70% | 1 4.35% | 0 0.00% | 5 21.74% |
| Smart cities | 6 26.09% | 5 21.74% | 6 26.09% | 1 4.35% | 1 4.35% | 0 0.00% | 4 17.39% |
| Industry 4.0 | 4 17.39% | 10 43.48% | 3 13.04% | 1 4.35% | 1 4.35% | 0 0.00% | 4 17.39% |

### 7.5. Additional Comments

In the survey, the students could also enter additional comments. More than a dozen respondents took advantage of this opportunity. Figure 9 outlines the meaning of the comments. In the comments, the students most often emphasized that they would like to learn more about AI, ML, IoT, and EC. It was also pointed out that these issues are rarely taught with adequate attention and detail (some issues are discussed too generally). Some of the students were also interested in the results of the survey.

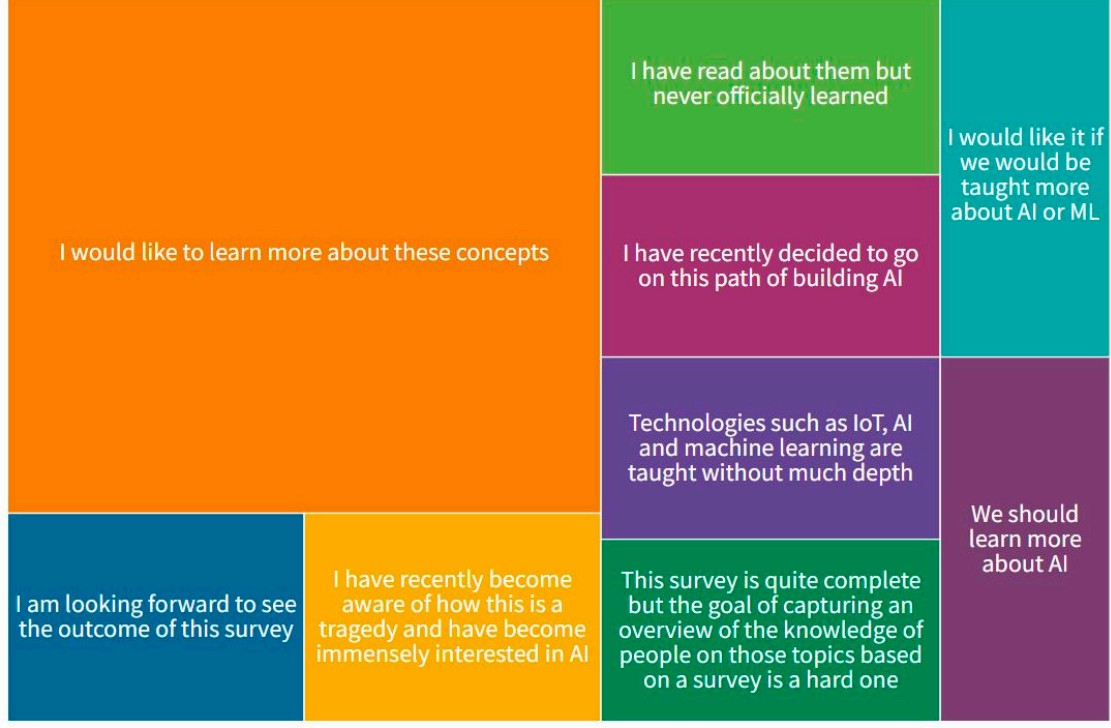

**Figure 9.** The meaning of the comments made by the students in the survey; the size of the boxes indicates the frequency of the answer.

## 8. Analysis and Discussion

In the work, the following research questions were discussed:

- RQ1: What knowledge and skills in the field of AI, IoT, and EC do the students possess and at what level?
- RQ2: What knowledge and skills in the field of AI, IoT, and EC are missing compared to the topics presented in literature?
- RQ3: To what extent do the students know how to apply AI, IoT, and EC in industrial problems solving?
- RQ4: How useful are different learning techniques for teaching AI, IoT, and EC?

### 8.1. RQ1: What Knowledge and Skills in the Field of AI, IoT, and EC Do the Students Possess and at What Level?

The first important result raised by the survey presented in the previous section is in what percentage students have ever learned about AI, IoT, and EC. A total of 54.4% of the students stating that they have learned about AI, compared to 24.8% regarding IoT, is not strange, and is probably due to the fact that the highest percentage of students participating in the survey come from IT Engineering (see Figure 3). However, what is really surprising is that only 2.32% of the students stated that they have learned about the three fields (Figure 4), considering the high complementarity among them in real scenarios. This could also be related to the other main highlight of this part of the survey, the low percentage of students who have learned about EC (only 4.1%). Although EC is a more recent topic compared to AI and IoT, it is a clear field of interest for many real application scenarios (including I4.0), a confirmed (current and future) key technology by all the economic forecasts, and is of course supported by a mature and growing research community. EC brings AI to another computation paradigm, moving AI and ML to where data generation and processing take place, making it more secure and fast. Moreover, in the IoT scenario, EC is complementary (and even essential) for faster, redundant, connectivity-agnostic IoT processing that is readily scalable. Thus, it is a main drawback to have so many students with AI or IoT skills but without any knowledge in EC, and universities have to quickly react to solve this important lack.

Regarding knowledge and skills in the specific field of AI, it is not surprising that the students stated a major knowledge in the general categories such as ML, deep learning, data mining, etc., than in the applied topics such as natural language processing or computer vision. Nevertheless, that more than 25% of the students stated that they did not know natural language processing and computer vision, that and more than 40% stated that they did not know cognitive computing at all, is quite disappointing and is probably due to the limited amount of hours related to AI in current curricula (the lack of hours of AI lectures was also claimed by the students in their comments). In particular, the survey identified reinforcement learning as the most unknown technique in the ML category, with 28% of the students stating that they did not know the topic at all, and deep learning as a full category with a current low level of knowledge in all of the identified issues, going from the most positive 22% of the students not knowing convolutional neural networks at all, to the most critical 50% of the students not knowing anything about generative adversarial networks and transformers. In both cases, reinforcement and deep learning, the lack of knowledge is probably due to the lack of curricula update with the latest technological trends, as both topics are quite recent compared to the rest of the ML techniques (also an issue highlighted by the students in their comments).

From another point of view, the declared knowledge in the field of data mining raised an interesting result related to the transdisciplinarity of studies. As described in the previous section, the knowledge in data mining is presented in the survey as divided into six phases of CRISP-DM, and the survey identified that students rated their knowledge as the worst in the first and last phases of CRISP-DM (business understanding and deployment), while the four internal phases of CRISP-DM that are related to data understanding and processing, modeling, and evaluation, were rated better. Indeed, these results indicate the

need to familiarize students with the business context of the analyzed data. If we want to form good data analysts, it is essential to provide students with not only technological knowledge on data processing and modeling, but also with business understanding and deployment background. This brings us to the broader discussion of to what extent new curricula should be more transdisciplinary, breaking the current verticals separating so much technology from business or other social sciences, if we want to form the professionals our society is currently demanding.

Finally, the survey also identified fuzzy logic as the least-known aspect of computation intelligence, with more than a third of the students stating that they did not know it at all. This may be due to the fact that fuzzy logic is not necessarily a technique that comes up often in typical software development; however, in our opinion, it is still important from an educational point of view for consolidating mathematical skills related to projects involving decision-taking and uncertainties. Regarding the students' programming knowledge for AI applications (Table 12), and in the software platforms used for AI implementation (Table 13), it can be noted that, for all the languages and tools, more than 50% of the students rated their knowledge in the three worst positions, that is, "not at all", "to a small extent", and "to some extent", and more than 70% of the students did not know at all tools such as Aitech Sphinx, Scilab, or SWI Prolog, and languages such as Lisp and Prolog. Also surprising was that 60% of the students stated that they did not know R language at all.

Regarding students' knowledge and skills in the specific field of IoT, the survey identified that about 95% of the students stated that they have at least a basic knowledge of general concepts of IoT and IoT application scenarios. However, again, important topics such as M2M industrial IoT protocols, searching for vulnerabilities, the distribution of computing processes in IoT networks, IoT maintenance, or cryptography, still showed 30% of the students to not have any knowledge about them at all. Furthermore, the level of knowledge in terms of the application of IoT in several contexts is analyzed (Table 20), obtaining lower results compared to the general topics of IoT included in the previous question. In all contexts, at least 23% of the students declared the lack of any knowledge of using IoT for market behavior, logistics, and deliveries—the most critical ones. Again, it can be observed that the transdisciplinarity of the courses (including business scenarios) is a major lack of the current curricula of these novel technologies. Finally, regarding the software used in IoT (Table 23), the survey detected that more than half of the students have not used AWS Lambda, which is very important to learn for serverless computing and event-driven programming. Moreover, MapReduce is not known at all by more than half of students, which is probably due to the presence of other better alternatives, but it is still surprising because, from an academic point of view, it is very valuable for demonstrating the underlying methods by which data are processed in all distributed systems.

Regarding students' knowledge and skills in the specific field of EC, something similar to IoT and AI occurred. The general concepts and applications seem to be well covered by the curricula, but when asking about more specific topics, such as privacy and security, scalability and reliability, or speed and efficiency, about 30% of the students stated that the did not know the topics at all or just to a small extent. This again may be due to the limited hours spent in teaching these subjects and should, indeed, be solved by providing more courses in these areas. In fact, these results are aligned with the even worse level of knowledge declared by the students regarding the technologies used in EC. 40% of them stated not knowing at all or having little knowledge in all the technologies presented in Table 25 (mobile EC, fog computing, service composition and service-oriented computing, etc.), and this even becomes nearly 50% in the case of Azure edge. Regarding algorithms and techniques for EC (Table 26), it is also worth mentioning that more than 50% of the students did not know containerization at all. Due to the limited number of students participating in this part of the survey (which is already a very significant result in itself), it may be difficult to extract general conclusions regarding EC, and this becomes even more critical regarding the use of hardware and software used in EC platforms, where it seems there is not a widely known tool.

### 8.2. RQ2: What Knowledge and Skills in the Field of AI, IoT, and EC Are Missing Compared to the Topics Presented in Literature?

As stated in the previous answer, one of the most surprising results of the survey is that only 4.1% of the students have learned about EC. The incredible interest that this new technology has brought in the academic world, as we can see in Figure 10, where the number of papers published per year inside the Scopus database is shown, makes this result even more astonishing. Moreover, the low level of knowledge of related technologies is quite worrying, given the number of papers about them. Some examples are given by [35], in which the authors present the advantages and applications of fog computing in an IoT scenario, and by [36], where it is described why Mobile EC can help the transfer from cloud-based computing to the more easily scalable EC paradigm.

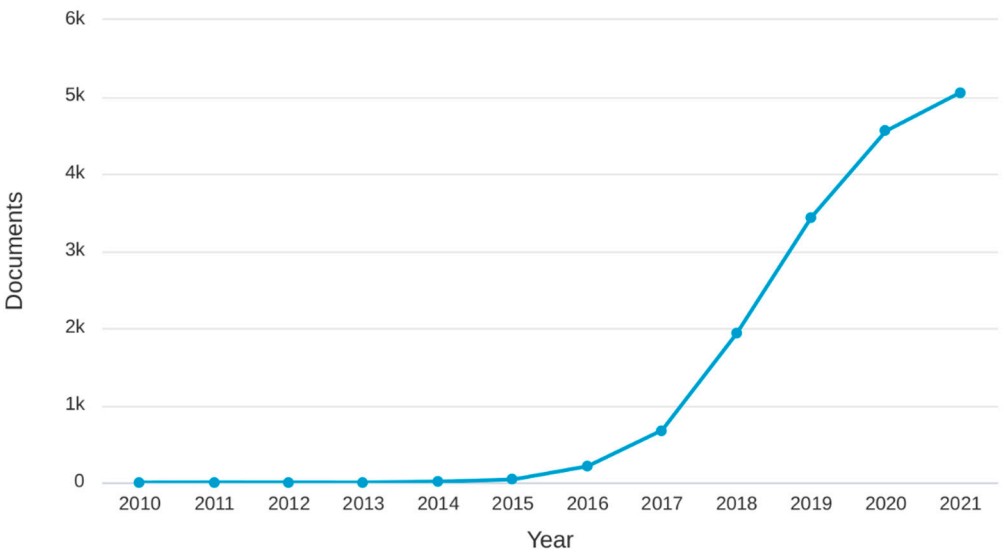

**Figure 10.** The number of papers on EC in Scopus.

Regarding AI, an interesting aspect is the low percentage of students who know more sophisticated ML architectures, such as convolutional neural networks and transformers, given the huge gain in performance they obtained with respect to older models [37,38] and the number of real-world applications they can solve [39–41].

From the IoT point of view, the survey shows a general ignorance about IoT applications and use cases, as well as the challenges that these applications are required to face. In particular, 30% of students who did not know or knew a small extent about privacy and security, scalability, reliability, speed, and efficiency clash with the various studies and solutions for these problems, well described in [42–44], respectively.

From a more practical perspective, the fact that the students highlighted the need for a more applied approach probably implies the lack of important soft skills required by the modern world of work. Students in fact manifest the importance and utility of problem-based learning, labs, and workshops, with the percentage of students who strongly believe in their usefulness ranging from 25% to more than 30%. Working on practical examples and case studies usually develops problem-solving, communication, and teamwork skills, whose importance for the future workers, especially after the fourth industrial revolution, is pointed out in a lot of academic works, for example, in [45–47].

### 8.3. RQ3: To What Extent Do the Students Know How to Apply AI, IoT, and EC in Industrial Problems Solving?

The first question that indirectly concerns RQ3 is related to knowledge of the data mining phases (see Table 7). As mentioned in RQ1, the students rated the knowledge of two phases of data mining (business understanding and deployment) as the worst.

However, unfortunately, these phases can be crucial in the context of AI implementation in the industry. The business understanding phase allows students to understand the business field and correctly locate the problem that is being solved in this field. In turn, deployment is to prepare the student to apply previously created data mining models in the business field. Gaps in knowledge in this area can cause problems when students want to move from academia to the world of industry and business.

The next two survey questions that may help answer RQ3 concern aspects of natural language processing and computer vision. Both of these issues deal with practical aspects, such as speech recognition or object localization and detection. These aspects relate to the many practical tasks that students in the industrial and business world can face, such as identifying defective products based on product photos or creating chatbots to communicate with customers. However, more than a third of the students are unfamiliar with any aspects of natural language processing (see Table 9). Only less than 10% of the students indicated that they know natural language processing to a great or very great extent. The situation is similar in the case of computer vision aspects (Table 10). A very small group of students indicated a high or very high level of knowledge of these issues. Such results may indicate a gap between theoretical issues and their practical application. For example, students may know how a deep neural network works, but may have a serious problem with its application, for example, to classify product images during a quality control on a production line. This gap should be filled by embedding in the curricula-relevant examples and references to industry issues that can be supported by natural language processing and computer vision.

In order to solve practical industrial problems, students should know some tools that facilitate this. Appropriate programming languages can be such a tool. One of the close-ended survey questions concerned the knowledge of selected programming languages in AI applications (see Table 12). In addition, in an open-ended question, students could indicate additional programming languages they already know. The students' answers show that the two programming languages that are best known to students are Python and MATLAB. Relatively high knowledge of these programming languages can be positive in the context of AI in industrial problem-solving applications. There are many examples of practical implementations and solutions in the scientific and business literature that are based on both MATLAB and Python. Therefore, developing students' skills in these languages seems to be absolutely justified, and the knowledge obtained by students may be useful to them on the labor market.

One of the survey questions is directly related to the students' knowledge of how to use AI in selected applications. This close-ended question lists several potential AI applications such as quality problems, predictive maintenance, or manufacturing process monitoring (Table 14). The question contributes significantly to RQ3. In addition, in one of the open-ended questions, the students could indicate other applications in which they used AI. Forty-four students used this opportunity and added some additional examples of AI applications (Table 15). They most often mentioned AI applications related to healthcare or medicine, game development, agriculture, chatbots, finance, and recommendation systems. The students' answers show that they have the best knowledge of how to apply AI to optimization problems. More than 60% of students had at least a basic knowledge of this subject. However, in the case of other applications, the number of students did not exceed 60%. The worst situation is in the case of supply chains management and manufacturing process monitoring, where approximately half of the students did not have any knowledge of how to use AI in these tasks. A similarly low level of knowledge concerns deliveries, scheduling problems, cognitive systems, and robots, where more than 40% of the students indicated a lack of knowledge. These insights may indicate the necessity to pay attention to the practical aspect of using the acquired knowledge in the field of AI, in particular in the areas mentioned above. Comparing the data on the applications of AI with the level of general students' knowledge about areas of AI (see RQ1), it can be concluded that the

students evaluate their general knowledge of the methods, techniques, and tools of AI better than the knowledge of their applications.

The students also mentioned the need for close contact with the industry in open-ended questions about what is missing in AI education. One of the most frequently heard opinions concerned too few real-life case studies, laboratory classes, and workshops. According to the students, the AI learning process should use contacts with the industry and professionals, especially in the form of internships, projects, and competitions. This shows that many students are aware of the benefits of contact with the industry, and that they would be eager to expand their AI competences in collaboration with the industry.

In order to determine to what extent the students know how to apply IoT in industrial problems solving, it is necessary to pay attention to three close-ended survey questions. The first one is directly related to RQ3. In this question, the students defined their knowledge of the use of IoT in various contexts. The answers to this question show that, among the contexts considered, the students had the best knowledge of robotics. Approximately 12% of the respondents indicated at least a "to a great extent" level of use of IoT in robotics (see Table 20). Moreover, for robotics, the "not at all" response rate was the smallest. Relatively similar to robotics, the students' responses were distributed in the context of data management, machine condition monitoring, or support decision making. Nevertheless, it would be wrong to conclude that the mentioned contexts of IoT are sufficiently well known to the students. Even in the case of the best-known context of robotics, there is still a lot of catching up to do, because almost a quarter of the students did not know this context of IoT at all, and a quarter knew it only to a small extent. An even lower level of knowledge can be seen in the contexts of "delivery" and "logistics", where more than 38% of the respondents indicated that they did not have any knowledge. These results show that only a few of the students theoretically know how to apply IoT.

The same observations can be drawn from the next question in which the knowledge of software or environments used in IoT was considered. These are the tools by which IoT can move from theory to practice. As it turns out, the level of use of software mentioned in the close-ended question is very low (see Table 23). This will certainly not be conducive to using IoT in industrial problems.

However, when looking at the first question in the IoT section of the survey, it can be assumed that the students knew the theory about the use of IoT much better than its practice. In this question, the students indicated the level of their knowledge, among other things, about IoT application scenarios. Only 5% of the students indicated that they do not have any knowledge of this topic and more than 20% assessed that they know the application scenarios of IoT at least to a great extent.

The problem of gaining knowledge and skills regarding the practical applications of IoT may be related to the fact that IoT is used in many different fields and problems. As the students pointed out in the open-ended question "what is the most difficult thing in learning IoT", the answer was quite often that IoT is a very broad concept and can cover many issues, problems, and examples of use. On the other hand, when speaking about the shortcomings in the IoT education process, the students pointed out that too little emphasis was put on solving real-word problems and application-driven education. It can be assumed that filling these gaps would contribute to acquiring significant competences of students in the field of industrial problem solving using IoT.

In the case of EC in industrial problem solving, the students' answers to three close-ended questions are especially significant. The most important of them is the question about the level of knowledge about selected EC applications (e.g., autonomous machines, augmented reality, or smart cities). None of the students indicated a very high level of knowledge and only a few students indicated a high or medium level of knowledge on EC applications (see Table 29). The vast majority of the students replied that they had no or only low-level knowledge of how to apply EC. These results indicate that future industrial workers may have serious difficulty in applying their knowledge to the practical

application of EC. It is hard to talk about solving industrial problems using EC if students do not have knowledge of the practical applications of EC.

In the context of solving industrial problems, the knowledge of technologies, algorithms, and techniques that are used in EC implementation is also significant. The next close-ended question raised the issue of technology related to the implementation of EC (see Table 25). Among the considered technologies, the best known seems to be container technology, and the least known is Azure edge. The students most often indicated a medium or low level of knowledge, while no one indicated a very high level of knowledge. The results of the next question, in which the students indicated the level of knowledge about the techniques and algorithms used in the EC implementation, are similar (Table 26). The highest number of responses concerned "to a small extent", "to some extent", and "to a moderate extent" variants. Summing up, due to the fact that about 20% of the respondents did not indicate any level of knowledge in EC technologies and algorithms, and at least a dozen or so percent of the respondents did not know these issues at all, it can be concluded that each considered technology and algorithm may be a potential need in the EC education process. The current level of students' knowledge cannot guarantee that they are sufficiently prepared to face the problems they may encounter as industrial workers.

### 8.4. RQ4: How Useful Are Different Learning Techniques for Teaching AI, IoT, and EC?

The survey's answers presented a relatively common pattern as regards the most useful learning techniques for all under-consideration technologies. Laboratory classes emerged as the most desirable learning technique among the students. Project-based learning and workshops are also high on the preference list of the respondents. On the other hand, even though lectures and e-learning appeared as the least preferable, they also have keen advocates. Such a finding directs us to draw conclusions in two directions.

First of all, the need for more practical teaching methods is demonstrated. Students' request for more applied educational techniques is more or less apparent throughout the entire survey. Besides, it is also indicated that the practical realization of some tasks (e.g., in a project) gives students the most knowledge and skills. Secondly, it is clear that the traditional teaching methods (such as lectures) are not wholly rejected as an option, but rather need reformation. We infer from these two conclusions the imperative need (as exhibited in students' answers) for creating an educational environment wherein several diverse learning techniques would be offered.

Monolithic teaching approaches seem to have no place in a modern educational program, let alone considering the fact that we discuss cutting-edge technologies, which rapidly evolve and incorporate new features while they are actually applicable in a wide variety of fields [48]. Students recognize the need for enhancing their theoretical background, especially regarding mathematical issues or having a vague perception of the logic behind these technology issues. Nevertheless, they underline the need for finding a balance between theory and practice; in other words, finding the golden ratio of building concrete and specialized awareness of a specific topic's theoretical background and gaining an experiential insight dealing with real problems. They are overwhelmed from being bombarded with the enormous theory that is not connected to real applications.

Therefore, it is essential to define a clear implementation strategy with regard to the educational process. We have to develop agile educational programs that meet the diverse preferences and the learning pace of every student. Blended learning seems to be the optimum solution, where different learning techniques are combined to obtain the best qualities of them. The most important finding is that we have to propose and adapt some practical aspects of the under-examination technologies. Learning-by-doing emerges as a demand. Laboratory exercises where students learn through the experimental process, workshops, and project implementation indeed serve this approach. What is more, work-based learning—in the sense of an internship in a highly specialized company—could also be beneficial for practical learning. Additionally, and directly related to the latter, the collaboration with companies and, in general, experienced professionals of the field

(e.g., seminars, demonstrations, etc.) would not only contribute to the practical aspect, but also would represent a reference point for keeping updated with the latest advancements. Learning factories for serving the teaching of the transition from conventional to I4.0-based systems [49] have been proven to play an important role in the development and realization of the I4.0 concept through the collaboration of researchers, companies, and academia. They can incorporate and test a wide range of I4.0 technologies through test-beds and proof-of-concept experiments, supporting the experimental validation of different artifacts, such as modeling and simulation frameworks [50].

On the other hand, online learning primarily avails if it can be used as a centralized place (e.g., a platform) where a student, at his/her own pace, could study utilizing extended open-source repositories providing access to good quality training material and resources (e-books, an environment to experiment with the projects, good quality assisting code, recommendations on projects according to someone's level of competence, provision of useful open-source software, active forum). The traditional learning such as lectures are valuable if they become more flexible (e.g., case studies with emphasis on real-problems solving, visualization of the key concepts of the theory, etc.). Last but not least, interactive learning environments, such as virtual, mixed, or augmented reality, would definitely enrich the educational process with new characteristics. Such an approach offers the benefit of practically experimenting with technologies without the need for establishing expensive infrastructures in universities. Simulations of simple or more complex IoT systems could facilitate the educational process by experimenting (changing parameters' values, examining the response of the system, training in the manipulation of robotic systems, etc.). Virtual reality environments combined with playful and gameful learning (gamification) [51] can boost and support creative student-centered teaching.

This survey spotlighted some lack of practical approaches to teaching the under-examination technologies. Respondents, even those declaring a better-than-moderate theoretical knowledge of one technology, showed decreased levels of competence regarding its practical application. Such a finding should not be underestimated in any circumstances if we desire to experience the full potential of I4.0 technologies on a large scale. Educating the students on the practical aspects of these technologies could eventually induce the narrowing of the existing gap between the knowledge transferred by academia and the demands of the technologically modern companies. The students' replies also highlight the need for embracing new and versatile teaching techniques from the universities.

## 9. Conclusions

In the last decade, we have witnessed the abrupt establishment of the so-called I4.0 era. The production model has been transformed, and more and more processes are implemented by smart machines. The technologies that realize such a revolution evolve rapidly, rendering the modern worker in the position of struggling to keep track. Three of the most pivotal technologies that characterize I4.0 are AI, IoT, and EC. The latter, and specifically, the depth and extent of teaching them in the European universities, were the subject of the research presented in the current paper and conducted via the students' participation in a survey. The ultimate goal was to examine the extent of the existence of a gap between the needs of companies and the knowledge and skills that students acquire at universities regarding these technologies.

The main finding of the research highlighted the fact that AI technologies and applications are by far more familiar to students, while EC is the technology with which the students are the least acquainted, being the most recently introduced term. This means that academia does not present the readiness to incorporate cutting-edge technologies in its curricula. On the other hand, when it comes to the practical aspect of them, things worsen. Students seem to have a much better theoretical background in comparison to their knowledge of how these technologies can be applied in real working environments. Another important conclusion is that the students' understanding of the modern business model is discouraging. Moreover, the research pointed out the need for adopting new

and versatile learning techniques that would serve students' request for introducing the practical aspect of the under-examination technologies in the educational procedure. Such a transformation would eventually narrow down the existing gap between academia and industry, as the future workers would be adequately prepared and trained to undertake and accomplish real industrial problems.

As with any other research work, we dealt with different work limitations. First of all, we tried to conduct a large-scale survey worldwide; nevertheless, the reluctance of participation limited our ambition. However, the total number of respondents is considered satisfactory. Furthermore, another concern is related to the objectivity of replies. Self-assessment of knowledge via a scale may sometimes become a disorienting factor. However, we do not think that the quality of the results is under dispute due to the fact that the results showed a pattern and cohesion. What is more, the participation in the survey of each country was not equal regarding the number of replies we got; consequently, it is under examination if such a fact reflects on the overall conclusions.

The results and conclusions of this survey can be exploited by all the interested parties. The target groups, indeed, are the universities, businesses and companies, and the industrial sector, but, of course, any other individual, public bodies, or decisionmakers could find this work useful in the process of policy-making and designing the future. The implications of our work can vary from delivering a fresh perspective of the educational needs that are aligned to modern companies' needs, to facilitating an individual's desire to pursue and design his own path for acquiring the most in-demand skills and competences in the labor market. Therefore, in the first place, we provide the raw material that can be utilized to conduct a large-scale transformation of the educational programs and curricula in academia in favor of the whole society.

Regarding future research, as academic representatives, we intend to promote the integration of other scientific fields (e.g., social sciences, business science) in the teaching of AI, IoT, and EC to serve an interdisciplinarity that is so crucial in the new era, whereby the citizen is provided with all the required skills and knowledge to be flexible in, and adaptable to, diverse and rapidly changing working environments. It would be of great interest to experiment and examine the degree of acceptance and penetration of such an expedition and the subsequent benefits that may occur. This research could proceed one step further by incorporating and examining the level of students' competence in other state-of-the-art technologies that are components of the I4.0 model, and therefore gaining a deeper insight into them. Last but not least, based on the experience we gained, we can refine the survey and use it as a prototype for relevant surveys.

**Author Contributions:** Conceptualization, D.S.; methodology, D.S.; validation, C.S., D.M., G.D. and Ł.P.; formal analysis, D.S.; investigation, D.S., G.D., M.M., Ł.P., A.C.-C., X.S.-B., L.P., C.S., D.A. and D.M.; resources, D.A., A.C.-C. and L.P.; writing—original draft preparation, D.S.; writing—review and editing, A.C.-C., X.S.-B., D.A., D.M., L.P., C.S. and Ł.P.; visualization, M.M., D.S., D.A. and Ł.P.; supervision, D.S. and Ł.P.; project administration, D.S.; funding acquisition, C.S. All authors have read and agreed to the published version of the manuscript.

**Funding:** This work has been partially funded by Programme Erasmus+, Knowledge Alliances, Application No 621639-EPP-1-2020-1-IT-EPPKA2-KA, PLANET4: Practical Learning of Artificial iNtelligence on the Edge for indusTry 4.0.

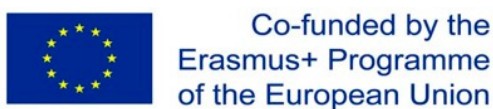

**Institutional Review Board Statement:** Not applicable.

**Informed Consent Statement:** Not applicable.

**Data Availability Statement:** Not applicable.

**Conflicts of Interest:** The authors declare no conflict of interest.

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
