# Peer review of "Plan and Develop Advanced Knowledge and Skills for Future Industrial Employees in the Field of Artificial Intelligence, Internet of Things and Edge Computing"

_sustainability, doi:10.3390/su14063312_

Round 1

Reviewer 1 Report

Excellent job,

They have material for two articles,

Consider this article to make two articles or summarize this one because it is too big,

However, consider considering reinforcing and clarifying the objective of the research (What is the objective of the study? What is the relevance of the study? What did the study bring us? What are the contributions?)

Is there no more recent investigation?

I suggest the following articles that have enriched the literature review

Rosário, A. T. (2021). The Background of articial intelligence applied to marketing, Academy of Strategic Management Journal, 20(6), 1-19, ISBN: 1939-6104, doi.org/1939-6104-20-S6-118

Raimundo, R. & Rosário. A. T., (2021). The impact of Artificial Intelligence on data system security: a literature review, sensors, 21(21), 7029, ISSN: 1424-8220, doi.org/10.3390/s21217029

Raimundo, R. & Rosário, A. T. (2022). Cybersecurity in the Internet of Things in Industrial Management, Applied Sciences, 12(3), 1598, ISSN:  2076-3417, doi.org/10.3390/app12031598

Methodologically, it could be clearer and more objective. I felt some difficulties in understanding the methodological process, I would have many difficulties in replicating the study.

Ponder an implications section

I hope I was helpful,

No other subject,

The hug

Reviewer 2 Report

Strengths

The article presents a unique research topic and scope related to planning and developing advanced knowledge and skills for future industrial employees in the field of artificial intelligence, the Internet of things, and edge computing.

The article presents interesting data and detailed analyzes.

The solutions to the research problem contained in the text complement the scientific literature in the scope of the present issues.

The solutions presented by the authors are interesting and broaden the scope of research related to planning and developing advanced knowledge and skills for future industrial employees in the field of artificial intelligence, the Internet of things, and edge computing.

The conclusions presented by the authors, based on the analysis of the literature are consistent with the evidence and arguments.

Weaknesses

There is no precisely defined purpose of the manuscript.

The significance of the research problem has not been presented in detail.

The research methodology has not been described in detail.

There are no references to the directions of further research.

Authors should:

The purpose of the article should be specified.

Define the essence and importance of the described problem.

The methodology should be completed in detail of the conducted empirical research.

 Lines for further research should be identified.

Round 2

Reviewer 1 Report

Bear,
Good work

Reviewer 2 Report

The new version of the manuscript contains mostly additions and corrections.